# Isotopic constraints on the role of hypohalous acids in sulfate aerosol formation in the remote marine boundary layer

Qianjie Chen[1], Lei Geng[1,*], Johan A. Schmidt[2], Zhouqing Xie[3], Hui Kang[3], Jordi Dachs[4], Jihong Cole-Dai[5], Andrew J. Schauer[6], Madeline G. Camp[7,**] and Becky Alexander[1]

[1] Department of Atmospheric Sciences, University of Washington, Seattle, WA, USA

[2] Department of Chemistry, University of Copenhagen, Copenhagen, Denmark

[3] Institute of Polar Environments, School of Earth and Space Sciences, University of Science and Technology of China, Hefei, Anhui, China

[4] Department of Environmental Chemistry, IDAEA-CSIC, IDAEA-CSIC, Barcelona, Catalunya, Spain

[5] Department of Chemistry and Biochemistry, South Dakota State University, Brookings, SD, USA

[6] Department of Earth and Space Sciences, University of Washington, Seattle, WA, USA

[7] Joint Institute for the Study of Atmosphere and Ocean, University of Washington, Seattle, WA, USA

*Now at* Univ. Grenoble-Alpes, LGGE, F-38000, Grenoble, France and CNRS, LGGE, F-38000, Grenoble, France

**Now at* DSG Solutions, LLC, Shoreline, WA, USA

*Correspondence to*: Becky Alexander (beckya@atmos.washington.edu)

**Abstract.** Sulfate is an important component of global atmospheric aerosol, and has partially compensated for greenhouse gas-induced warming during the industrial period. The magnitude of direct and indirect radiative forcing of aerosols since preindustrial time is a large uncertainty in climate models, which has been attributed largely to uncertainties in the preindustrial environment. Here, we report observations of the oxygen isotopic composition ($\Delta^{17}O$) of sulfate aerosol collected in the remote marine boundary layer (MBL) in spring and summer in order to evaluate sulfate production mechanisms in pristine-like environments. Model-aided analysis of the observations suggests that 33-50% of sulfate in the MBL is formed via oxidation by hypohalous acids (HOX=HOBr+HOCl), a production mechanism typically excluded in large scale models due to uncertainties in the reaction rates, which are due mainly to uncertainties in reactive halogen concentrations. Based on the estimated fraction of sulfate formed via HOX oxidation, we further estimate that daily-averaged HOX mixing ratios on the order of 0.01-0.1 parts per trillion (ppt=pmol/mol) in the remote MBL during spring and summer are sufficient to explain the observations.

## 1 Introduction

Large uncertainties in estimates of aerosol radiative forcing, especially those induced by sulfate aerosol and its interaction with clouds, significantly impede the progress of constraining the magnitude of anthropogenic radiative forcing since preindustrial times (Myhre et al., 2013). The aerosol radiative forcing uncertainties are attributed in large part to our poor understanding of the abundance of natural aerosols, especially sulfate aerosol in the marine boundary layer (MBL) (Carslaw et al., 2013) that is mainly produced from the oxidation of dimethylsulfide (DMS) emitted from oceanic phytoplankton (Bates et al., 1992). The radiative effects of sulfate involve scattering of solar radiation and modification of cloud properties (Haywood and Boucher, 2000). Determining the magnitude of these radiative effects requires in part understanding of sulfate formation mechanisms. Only sulfate formed via gas-phase oxidation can nucleate new particles (Kerminen et al., 2010; Kulmala et al., 2000) with implications for particle number density. Sulfate formed in the aqueous phase impacts particle growth rates in clouds with implications for aerosol size distribution (Lelieveld and Heintzenberg, 1992).

In the MBL, due to the high solubility and fast aqueous-phase oxidation of $SO_2$, the main sulfate production mechanisms are thought to be in-cloud oxidation of dissolved $SO_2$ ($S(IV) = SO_2 \cdot H_2O + HSO_3^- + SO_3^{2-}$) by hydrogen peroxide ($H_2O_2$) and ozone ($O_3$) (Faloona, 2009; Alexander et al., 2012). In addition to sulfate formation in clouds, MBL sulfate formation can occur via oxidation of $SO_2$ by OH in the gas phase (Stockwell and Calvert, 1983) and on the surface of sea-salt aerosols in the presence of $O_3$ (Sievering et al., 2004; Alexander et al., 2005). Other sulfate production mechanisms that are important in specific environments, such as metal-catalyzed oxidation of S(IV) by $O_2$ (Alexander et al., 2009; Harris et al., 2013; 2014) and gas-phase oxidation of $SO_2$ by Criegee intermediates (Mauldin III et al., 2012; Pierce et al., 2013), are thought to be minor in the MBL.

Modeling studies by Vogt et al. (1996) attributed a large part (60%) of aqueous-phase sulfate production in the MBL to oxidation of S(IV) by hypohalous acids (HOX = HOBr + HOCl); Further, von Glasow et al. (2002) evaluated the contribution of HOX to sulfate formation in both the cloud-free and cloudy MBL with a numerical one-dimensional model. Under both cloud-free and cloudy MBL conditions, about 30% of sulfate was formed via oxidation of S(IV) by HOX in the aqueous phase. Despite the potentially important role of HOX in sulfate formation in the MBL, the "S(IV)+HOX" reaction has not been included in most large-scale models of sulfur chemistry due to large uncertainties in (1) rate constants for reactions between HOX and $HSO_3^-$, (2) the Henry's law constant for HOCl ($H_{HOCl}$) and HOBr ($H_{HOBr}$), and (3) concentrations of HOX in the MBL.

Laboratory experiments demonstrate that reactions between HOX and $SO_3^{2-}$ occur via nucleophilic attack of $SO_3^{2-}$ onto the X atom of HOX (X = Br or Cl), followed by rapid hydrolysis of $XSO_3^-$ (Yiin and Margerum, 1988; Fogelman et al., 1989; Troy and Margerum, 1991):

$$HOX + SO_3^{2-} \rightarrow OH^- + XSO_3^- \tag{1}$$

$$XSO_3^- + H_2O \rightarrow SO_4^{2-} + X^- + 2H^+ \tag{2}$$

The rate constants for the "$HOBr+SO_3^{2-}$" ($k_{HOBr+SO_3^{2-}}$) and "$HOCl+SO_3^{2-}$" ($k_{HOCl+SO_3^{2-}}$) reactions are $5\times10^9$ $M^{-1}s^{-1}$ and $7.6\times10^8$ $M^{-1}s^{-1}$, respectively (Fogelman et al.,1989; Troy and Margerum, 1991); In addition, Liu (2000) suggested that the reaction of HOBr with $HSO_3^-$ follows a similar pathway as with $SO_3^{2-}$, but with a lower reaction rate constant due to reduced nucleophilicity of $HSO_3^-$ compared to $SO_3^{2-}$:

$$HOBr + HSO_3^- \rightarrow H_2O + BrSO_3^- \tag{3}$$

$$BrSO_3^- + H_2O \rightarrow SO_4^{2-} + Br^- + 2H^+ \tag{4}$$

They were unable to obtain $k_{HOBr+HSO_3^-}$ from their laboratory experiments due to interference from the unavoidable reaction of HOBr with $Br^-$ in acidic solution. Based on their laboratory results, they suggested an upper limit for $k_{HOBr+HSO_3^-}$ ($< 3.2\times10^9 M^{-1}s^{-1}$). To our best knowledge, there is no laboratory experiment that determined the reaction rate constant of HOCl with $HSO_3^-$ ($k_{HOCl+HSO_3^-}$). It is reasonable to assume that the reaction of HOCl with $HSO_3^-$ follows a similar pathway as the reaction of HOBr with $HSO_3^-$ (Eqs. 3-4). Lack of knowledge of $k_{HOBr+HSO_3^-}$ and $k_{HOCl+HSO_3^-}$ leads to large uncertainties in calculations of the sulfate formation rate from HOX because $HSO_3^-$ is the dominant S(IV) species (>93%) in clouds at typical cloud pH between 3 and 6 (Faloona, 2009).

Laboratory measurements of $H_{HOCl}$ range from 470 to 910 M atm$^{-1}$ (Holzwarth et al., 1984; Hanson and Ravishankara, 1991; Blatchley III et al., 1992). Based on the aforementioned laboratory results, Huthwelker et al. (1995) suggested an expression for $H_{HOCl}$ as a function of $H_2SO_4$ concentration and temperature. By assuming a temperature of 298.15 K and pure water, Sander et al. (2006) suggested $H_{HOCl} \approx 650$ M atm$^{-1}$. Estimates of $H_{HOBr}$ show a much larger range from 93 to 6100 M atm$^{-1}$ (McCoy et al., 1990; Blatchley III et al., 1992; Vogt et al., 1996; Frenzel et al., 1998; Sander et al., 2006; 2015). $H_{HOBr}$ = 93 M atm$^{-1}$ was assumed in the modeling studies by Vogt et al. (1996) and von Glasow et al. (2002), who simply estimated $H_{HOBr}$ as 10% of the solubility constant of HOCl at 293 K from Huthwelker et al. (1995). Frenzel et al. (1998) estimated $H_{HOBr}$ to be 6100 M atm$^{-1}$ using the Gibbs free energy of HOBr. Blatchley III et al. (1992) estimated $H_{HOBr}$ to be at least twice the Henry's law constant of HOCl that was measured in their laboratory experiments. Based on this relationship, Sander et al. (2006) extrapolated $H_{HOBr}$ to be $\geq$ 1300 M atm$^{-1}$ using $H_{HOCl}$ from Huthwelker et al. (1995). Only McCoy et al. (1990) measured $H_{HOBr}$ from laboratory experiments and reported $H_{HOBr}$ to be about 1900 M atm$^{-1}$.

Observations of HOX concentrations in the troposphere are sparse. Liao et al. (2012) made the first direct observation of HOBr in Alaska in spring 2009, and reported average daytime surface mixing ratios of about 10 ppt, consistent with the active bromine (HOBr+Br$_2$) mixing ratios measured by Neuman et al. (2010). HOBr mixing ratios were below their detection limit of 2 ppt at night (Liao et al., 2012). A recent aircraft campaign showed a daily-averaged HOBr mixing ratio of about 2 ppt along its flight track from 300 m to about 8000 m over the tropical Western Pacific during January and

February 2014, with very small vertical gradients (Le Breton et al., 2016). The detection limit of HOBr in Le Breton et al. (2016) was about 0.2 ppt. The only direct observation of HOCl mixing ratio was made over the eastern tropical Atlantic at the surface during June 2009 and a large range from <5 ppt to 173 ppt was reported (Lawler et al., 2011). The detection limit of HOCl in Lawler et al. (2011) was 5 ppt.

The $\Delta^{17}O$ ($\approx \delta^{17}O - 0.52\delta^{18}O$) of sulfate is solely dependent upon the relative importance of the oxidants involved in its formation (Savarino et al., 2000), and thus provides an observational constraint for sulfate formation pathways (Lee and Thiemens, 2001; Lee et al., 2001; Jenkins and Bao, 2006; McCabe et al., 2006; Patris et al., 2007; Dominguez et al., 2008; Alexander et al., 2005; 2009; 2012). $\delta^{17}O$ or $\delta^{18}O$ is expressed as:

$$\delta^{x}O = \frac{R^{x}_{SA}}{R^{x}_{VSMOW}} - 1 \tag{5}$$

where $R^{x}_{SA}$ is the $^{x}O/^{16}O$ ratio of the sample, $R^{x}_{VSMOW}$ is the same ratio of Vienna Standard Mean Ocean Water (VSMOW) (Gonfiantini et al., 1993), and $x$=17 or 18. The $\Delta^{17}O$ value is expressed in unit of per mill (‰). Table 1 lists the $\Delta^{17}O$ of sulfate formed via different pathways. $\Delta^{17}O$ of sulfate produced from OH, $H_2O_2$, and metal-catalyzed $O_2$ oxidation pathways are 0 ‰ (Dubey et al., 1997; Lyons, 2001), 0.7 ‰ (Savarino and Thiemens, 1999), and -0.09 ‰ (Barkan and Luz, 2005), respectively, which were discussed in detail in Alexander et al. (2005; 2009) and Sofen et al. (2011) and will not be repeated here. Primary anthropogenic sulfate has a $\Delta^{17}O$ of 0 ‰ (Lee et al., 2002). Sulfate produced from $O_3$ oxidation has a $\Delta^{17}O$ of 6.5 ‰, assuming $\Delta^{17}O$ ($O_3$) = 26 ‰ (Vicars and Savarino, 2014). $\Delta^{17}O$ of sulfate produced from HOX oxidation has not been directly determined from laboratory experiments. Since HOX promotes sulfate formation via "$SO_3^{2-}$+HOX" reactions by adding one oxygen atom from $H_2O$ to sulfate instead of transferring its own oxygen atom (Fogelman et al., 1989; Troy and Margerum, 1991; Yiin and Margerum, 1988), the $\Delta^{17}O$ of sulfate produced from "$SO_3^{2-}$+HOX" reactions is expected to have the same $\Delta^{17}O$ as water (0 ‰) (Gonfiantini et al., 1993). Liu (2000) suggests the reaction of HOBr with $HSO_3^-$ follows a similar pathway as with $SO_3^{2-}$ (Eqs. 1-2), resulting in $\Delta^{17}O$ of 0 ‰ for sulfate produced via this reaction. We assume that the reaction of HOCl with $HSO_3^-$ follows a similar pathway as the reaction of HOBr with $HSO_3^-$ (Eqs. 3-4) and produces sulfate with $\Delta^{17}O$ of 0 ‰. Based on these mechanistic studies, the $\Delta^{17}O$ of sulfate produced from HOX oxidation is expected to be 0 ‰.

Here, we report observations of $\Delta^{17}O$ of sulfate in atmospheric aerosols collected over a large spatial domain in the remote southern hemisphere MBL during spring and summer. We use these observations, combined with a global chemical transport model, to estimate the role of HOX in sulfate formation in the MBL.

**2 Sampling and measurements**

Aerosol samples were collected on quartz fiber filters (Whatman) using high volume air samplers located at the front of the ships from two ship cruises: (1) "Malaspina" as part of the Malaspina Circumnavigation Campaign on board of RV

Hespérides (González-Gaya et al., 2014), and (2) "Xue-Long" as part of the 28th China Antarctic Research Expedition supported by the the Program of China Polar Environment Investigation and Assessment (Project No. CHINARE2011–2015) on board of the Xue-Long icebreaker. The quartz filters were pre-combusted at 450 °C for 8 hours and kept wrapped in aluminum foil and plastic ziplock bags before use. Most Xue-Long filters were changed every 48 hours while most Malaspina filters were changed every 24 hours. The sampler was connected to a wind direction vane to avoid contamination from the ship exhaust. After sampling, filters were kept wrapped in aluminum foil and plastic ziplock bags at -20 °C. Blank filters were processed as field samples. One-quarter of each Xue-Long filter and 1/8 of each Malaspina filter were sent to the University of Washington for isotope and concentration measurements. Our samples from the Malaspina campaign cover the track from Cádiz, Spain to Sydney, Australia via Rio de Janeiro, Brazil, Cape Town, South Africa and Perth, Australia between January 02 and March 23, 2011. The Xue-Long campaign starts from Shanghai China on November 4, 2011, traveling through West Australia, Zhong Shan station, Antartica, South Argentina and back to Shanghai, China following the original route, ending on April 5, 2012. Figure 1 shows the 5-day back trajectories calculated from the NOAA HYSPLIT model for all sampling locations (http://ready.arl.noaa.gov/HYSPLIT.php), which gives a broad picture of the origins of air parcels along our sampling track. Most of the air parcels arriving at our sampling locations were over the ocean for the previous 5 days ($\approx$lifetime of sulfate (Chin et al., 2000)), which suggests that the sulfate collected was mainly formed in the MBL. However, observations of polycyclic aromatic hydrocarbons (PAHs) during the Malaspina campaign suggest that samples collected over the subtropical Indian Ocean and Atlantic Ocean might have continental influence (González-Gaya et al., 2014).

Aerosol ion concentrations ($SO_4^{2-}$, $NO_3^-$, $Cl^-$, $NH_4^+$, $Na^+$, $K^+$, $Mg^{2+}$, $Ca^{2+}$) for filter samples from the Malaspina and Xue-Long campaigns were measured at South Dakota State University (USA) and University of Science and Technology of China (China), respectively. The analytical procedures for measurement of anions and cations using ion chromatography have been presented elsewhere (Cole-Dai et al., 1995; Jauhiainen et al., 1999). Typical instrumental analytical precision for all ions is better than 10% RSD (relative standard deviation) at the $\mu$g $l^{-1}$ level. Bromide aerosol concentrations ($[Br^-]$) were also measured for the Xue-Long samples (Supplementary data). There is no relationship between observed $Br^-$ concentration and $\Delta^{17}O(nssSO_4^{2-})$ nor our calculated $[HOX]_g$ (not shown) because factors such as aerosol pH, sunlight and oxidants play an important role in the acid-catalyzed formation of reactive halogens and removal of HOBr (Fickert et al., 1999; Schmidt et al., 2016). Similarly, there is no relationship between $[Br^-]$ and HOBr mixing ratios in the global modeling study by Schmidt et al. (2016) (not shown). Thus,,$[Br^-]$ alone is not a good proxy for the "HOBr + S(IV)" reaction.

In the remote MBL, total sulfate consists of sea-salt sulfate ($ssSO_4^{2-}$) and non-sea-salt sulfate ($nssSO_4^{2-}$). $ssSO_4^{2-}$ refers to primary sulfate emitted directly from sea water via the bursting of bubbles while $nssSO_4^{2-}$ refers to secondary sulfate produced from oxidation of $SO_2$. For the Xue-Long samples, the $nssSO_4^{2-}$ fraction ($f_{nss}$) was calculated using the mass ratio of ($ssSO_4^{2-}/Na^+$) = 0.252 g/g in seawater (Millero et al., 2007). For the Malaspina samples, due to a sodium blank in the

quartz fiber filters, we calculated $f_{nss}$ using the mass ratio of $(ssSO_4^{2-}/Mg^{2+}) = 2.115$ g/g in seawater (Millero et al., 2008). Only samples with $f_{nss}$ larger than 30% were used in this study, to minimize the effect of uncertainty in the $ssSO_4^{2-}$ fraction on calculations of $\Delta^{17}O(nssSO_4^{2-})$. In the end, 25 Malaspina and 42 Xue-Long samples were used in this study, for a total of 67 (out of 91) samples. The averaged $f_{nss}$ is 0.57±0.21 (1σ) and 0.74±0.19 (1σ) for the 25 Malaspina and 42 Xue-Long samples, respectively.

The samples are divided into four categories (Fig. 2): (I) Southern Ocean, (II) Antarctic coast, (III) subtropical MBL and (IV) tropical coasts, based on their geographical location. The number of samples is 13, 18, 19 and 17 for Category I, II, III and IV, respectively.

$\Delta^{17}O$ of sulfate on the aerosol filter samples were measured using the pyrolysis method described in detail in Geng et al. (2013). Briefly, the sulfate on the filters was first dissolved in 18 MΩ water, purified using ion chromatography, and converted to $Na_2SO_4$ using ion exchange resin (AG 50 W-X8, 100–200 mesh, H$^+$ form, Bio-Rad, Hercules, CA, USA). 30% $H_2O_2$ solution was added to remove organics, and $Na_2SO_4$ was then converted to $Ag_2SO_4$ using the ion exchange resin. The $Ag_2SO_4$ was dried to a solid in a quartz cup and each sample was placed in a zero-blank autosampler attached to the continuous-flow inlet of the isotope ratio mass spectrometer (IRMS). Each $Ag_2SO_4$ sample was individually dropped into a furnace (1000°C) under a continuous flow of helium (He) where it is pyrolyzed to form $Ag(s) + SO_2(g) + O_2(g)$. The byproduct $Ag(s)$ condenses on the walls of the quartz pyrolysis tube, while the byproduct $SO_2(g)$ was removed from the He flow with a cryogenic trap at liquid nitrogen temperature ($\approx 77$ K). The remaining product $O_2(g)$ is carried along the He flow to the IRMS, where the abundance of $^{16}O$, $^{17}O$, and $^{18}O$ in $O_2$ was measured, and from which $\Delta^{17}O$ was calculated. 54 samples were measured in triplicate, 9 samples were measured in duplicate, and 4 samples were measured once. The precision of $\Delta^{17}O$ is typically better than ±0.3‰ based on replicate analysis of standards. The $\Delta^{17}O$ obtained from IRMS is the $\Delta^{17}O$ of total sulfate on the aerosol samples ($\Delta^{17}O(SO_4^{2-})$). $\Delta^{17}O(nssSO_4^{2-})$ was calculated by dividing $\Delta^{17}O(SO_4^{2-})$ by $f_{nss}$, as $\Delta^{17}O(ssSO_4^{2-}) = 0$ ‰ (Bao et al., 2000).

## 3 GEOS-Chem model

We use v9-02 of the GEOS-Chem global 3-D model (http://www.geos-chem.org/) of coupled oxidant-aerosol chemistry (Park et al., 2004) to simulate atmospheric sulfur chemistry and interpret our $\Delta^{17}O(nssSO_4^{2-})$ observations. The model is driven by assimilated meteorological data from the NASA Goddard Earth Observing System (GEOS-5, http://gmao.gsfc.nasa.gov), which has a temporal resolution of 6 hours, with 3 hours for surface quantities and mixing depths. Model simulations for the analysis of the cruise data were performed at 2°x2.5° horizontal resolution and 47 vertical levels up to 0.01 hPa using GEOS-5 meteorology corresponding to the timing of sample collection, after spinning up the model for one year.

The sulfate simulations were conducted in aerosol-only mode that used archived monthly mean OH, $NO_3$, $O_3$ and total nitrate concentrations and production and loss rates for $H_2O_2$ from the full-chemistry simulation as described in Park et al. (2004). A diurnal variation as a function of solar zenith angle is applied to OH concentrations and photolytic loss rates of $H_2O_2$ in the model. $NO_3$ is set to be zero during daytime. Sulfate in the model was produced from gas-phase oxidation of $SO_2(g)$ by OH, aqueous-phase oxidation of S(IV) by $H_2O_2$, $O_3$, and metal-catalyzed $O_2$ (Alexander et al., 2009), and

heterogeneous oxidation on sea-salt aerosols by $O_3$ (Alexander et al., 2005). The parameterization of the metal-catalyzed S(IV) oxidation pathway is described in detail in Alexander et al. (2009). The trace metals included are Fe and Mn, whose oxidation states Fe(III) and Mn(II) catalyze S(IV) oxidation. Soil-derived Fe ($[Fe]_{soil}$) is scaled to modeled dust concentration as 3.5% of total dust mass while soil-derived Mn ($[Mn]_{soil}$) is a factor of 50 lower than $[Fe]_{soil}$. Anthropogenic Mn ($[Mn]_{anthro}$) is scaled as 1/300 of primary sulfate concentration while anthropogenic Fe ($[Fe]_{anthro}$) is 10 times that of $[Mn]_{anthro}$. We

assume that 50% of Mn is dissolved in cloud water as Mn(II) oxidation state. For Fe, we assume that 10% of $[Fe]_{anthro}$ and 1% of $[Fe]_{soil}$ is dissolved in cloud water. 10% of the dissolved Fe is in Fe(III) oxidation state during daytime and 90% at night. Primary anthropogenic emissions of sulfate are 3.5% of total anthropogenic sulfur emissions in Europe, an average of 1.5% in North America and 2.1% elsewhere. The anthropogenic emission inventories used in this study is the global emission inventory EDGAR v3 (Olivier et al., 2001), supplemented by regional inventories such as STREETS (Streets et al., 2006),

EMEP (Vestreng and Klein, 2002), and NEI2005 (Van Donkelaar et al., 2015). The oceanic DMS inventory is from Kettle et al. (1999). Sulfate formed from each oxidation pathway was treated as a separate "tracer" in the model with a corresponding $\Delta^{17}O$ value as shown in Table 1. Primary anthropogenic sulfate has a $\Delta^{17}O$ of 0 ‰ (Lee et al., 2002). The model calculates $\Delta^{17}O$ of bulk sulfate in the grid box ($\Delta^{17}O_{mod}(nssSO_4^{2-})$) based on the relative importance of each sulfate production mechanism for total sulfate abundance. This is then compared to the $\Delta^{17}O$ measurement of bulk sulfate collected on aerosol

filters ($\Delta^{17}O_{obs}(nssSO_4^{2-})$) to investigate sulfate formation mechanisms in the MBL (Sect. 4.2).

For pH-dependent S(IV) partitioning, bulk cloud water pH is calculated as described in Alexander et al. (2012). Large-scale models such as GEOS-Chem calculate the average chemistry of bulk cloud water rather than the chemistry of individual cloud droplets. This approach has been shown to significantly underestimate sulfate formation via oxidation of $SO_3^{2-}$ by $O_3$ by underestimating the fraction of S(IV) present as $SO_3^{2-}$ (Hegg et al., 1992; O'Dowd et al., 2000; Roelofs, 1993;

Yuen et al., 1996; Fahey and Pandis, 2003). We use the Fahey and Pandis (2003) algorithm and the Yuen et al. (1996) parameterization in GEOS-Chem to account for the effect of heterogeneity in cloud drop pH on S(IV) partitioning as described in Alexander et al. (2012).

# 4 Results

## 4.1 Observations of $\Delta^{17}O(nssSO_4^{2-})$ and $nssSO_4^{2-}$ concentration

Except for one sample near the coast of China with a relatively high concentration of $nssSO_4^{2-}$ ($[SO_4^{2-}]_{nss}$) (7.4 µg m$^{-3}$), the $[SO_4^{2-}]_{nss}$ observations vary from 0.2 to 3.5 µg m$^{-3}$, with an average of 1.2±0.8 (1σ) µg m$^{-3}$. Averaged $[SO_4^{2-}]_{nss}$ is 1.4±0.8 (1σ) µg m$^{-3}$, 1.4±0.7 (1σ) µg m$^{-3}$, 0.9±0.5 (1σ) µg m$^{-3}$ and 1.3±0.9 (1σ) µg m$^{-3}$ for samples in Category I, II, III and IV, respectively (Table 2). A latitudinal gradient of $[SO_4^{2-}]_{nss}$ is found in our data, where averaged $[SO_4^{2-}]_{nss}$ between 50°S and 70°S is 50% higher than samples between 20°S and 40°S (1.5 *versus* 1.0 µg m$^{-3}$). The difference is significant at the 95% confidence level.

Figure 2 shows the observations of $\Delta^{17}O(nssSO_4^{2-})$ ($\Delta^{17}O_{obs}(nssSO_4^{2-})$) and $nssSO_4^{2-}$ concentration ($[SO_4^{2-}]_{nss}$) at each sampling location. $\Delta^{17}O_{obs}(nssSO_4^{2-})$ values range from 0.0 to 1.6 ‰, with an average of 0.7±0.4 (1σ) ‰. Averaged $\Delta^{17}O_{obs}(nssSO_4^{2-})$ (Table 2) is 0.5±0.3 (1σ) ‰, 0.7±0.4 (1σ) ‰, 0.8±0.4 (1σ) ‰ and 0.8±0.4 (1σ) ‰ for samples in Category I, II, III and IV, respectively. The analytical error in $\Delta^{17}O_{obs}(nssSO_4^{2-})$ is estimated by calculating the standard deviation (1 σ) of the multiple measurements of each sample, which range from ±0.0-0.4 ‰ with an average of ±0.1 ‰. Though $\Delta^{17}O_{obs}(nssSO_4^{2-})$ in Category I (over Southern Ocean) is slightly lower than those in other categories, the < 0.3 ‰ difference between each category is generally smaller than the measurement uncertainty estimated from replicate analysis of standards.

Only sulfate formed via $H_2O_2$ and $O_3$ oxidation has a positive $\Delta^{17}O(nssSO_4^{2-})$, with $H_2O_2$ oxidation leading to $\Delta^{17}O(nssSO_4^{2-})=0.7‰$ and $O_3$ oxidation leading to $\Delta^{17}O(nssSO_4^{2-}) = 6.5$ ‰. We can calculate the maximum contribution from "S(IV)+$O_3$" ($f_{O3,max}$) for each sulfate sample by assuming no contribution from $H_2O_2$ (i.e. all of the aqueous-phase S(IV) oxidation occurs via $O_3$ oxidation):

$$f_{O3,max} = \frac{\Delta^{17}O_{obs}(nssSO_4^{2-})}{\Delta^{17}O(nssSO_4^{2-})_{O3}} \tag{6}$$

where $\Delta^{17}O(nssSO_4^{2-})_{O3} = 6.5$ ‰. This yields $f_{O3,max}$ ranging from 0.00 to 0.26 with an average of 0.12±0.06 (1σ) for all samples. Averaged $f_{O3,max}$ is 0.08±0.05 (1σ), 0.11±0.06 (1σ), 0.13±0.07 (1σ) and 0.13±0.06 (1σ) for samples in Category I, II, III and IV, respectively (Table 2). For samples with $\Delta^{17}O_{obs}(nssSO_4^{2-})$ larger than 0.7 ‰ (36 samples), we can calculate the minimum $O_3$ contribution ($f_{O3,min}$) by assuming that $H_2O_2$ is the only other oxidation pathway (i.e., no significant contribution from OH and HOX oxidation):

$$f_{O3,min} = \frac{\Delta^{17}O_{obs}(nssSO_4^{2-}) - \Delta^{17}O(nssSO_4^{2-})_{H2O2}}{\Delta^{17}O(nssSO_4^{2-})_{O3} - \Delta^{17}O(nssSO_4^{2-})_{H2O2}} \tag{7}$$

where $\Delta^{17}O(nssSO_4^{2-})_{H2O2} = 0.7$ ‰. For samples with $\Delta^{17}O_{obs}(nssSO_4^{2-})$ smaller than 0.7 ‰ (31 samples), $f_{O3,min}$ is 0. We obtain $f_{O3,min}$ ranging from 0.00 to 0.16 with an average of 0.03±0.04 (1σ) among all samples. Averaged $f_{O3,min}$ is 0.01±0.02 (1σ), 0.03±0.04 (1σ), 0.04±0.05 (1σ) and 0.03±0.04 (1σ) for samples in Category I, II, III and IV, respectively (Table 2).

## 4.2 Comparison of modeled *versus* observed $\Delta^{17}O(nssSO_4^{2-})$

Figure 3a shows the comparison between modeled and observed $\Delta^{17}O(nssSO_4^{2-})$ ($\Delta^{17}O_{mod}(nssSO_4^{2-})$ vs. $\Delta^{17}O_{obs}(nssSO_4^{2-})$) for the standard model run (as described in section 3). $\Delta^{17}O_{mod}(nssSO_4^{2-})$ represents the daily mean in the first vertical model level (below ≈100 m) at each of our sampling locations. The range of $\Delta^{17}O_{mod}(nssSO_4^{2-})$ is 0.7~5.6 ‰, overestimating the observations on average by a factor of 2.5. Averaged $\Delta^{17}O_{mod}(nssSO_4^{2-})$ is 3.6±1.1 (1σ) ‰, 1.1±0.3 (1σ) ‰, 1.9±1.0 (1σ) ‰ and 1.2±0.4 (1σ) ‰ for samples in Category I, II, III and IV, respectively. $\Delta^{17}O_{mod}(nssSO_4^{2-})$ overestimates $\Delta^{17}O_{obs}(nssSO_4^{2-})$ in all categories (Table 2). The discrepancy between $\Delta^{17}O_{mod}(nssSO_4^{2-})$ and $\Delta^{17}O_{obs}(nssSO_4^{2-})$ is most evident for samples in Category I (Southern Ocean), for which the model predicts 48% of sulfate is formed via $O_3$ pathway, compared to 1~8% estimated from the observations alone.

The model calculated fractional contributions to the sulfate burden from each oxidant, averaged over all samples, is $f_{mod,OH}$=0.20±0.14 (1σ), $f_{mod,H2O2}$=0.57±0.15 (1σ), $f_{mod,O3}$=0.20±0.19 (1σ) and $f_{mod,het}$ =0.02±0.03 (1σ), where $f_{mod,OH}$, $f_{mod,H2O2}$, $f_{mod,O3}$ and $f_{mod,het}$ represents the fractional contribution of gas-phase OH oxidation, in-cloud $H_2O_2$ oxidation, in-cloud $O_3$ oxidation and heterogeneous oxidation by $O_3$ on the surface of sea salt aerosol to the total sulfate burden, respectively. The corresponding fractional contributions for samples in different categories are shown in Table 2. $f_{mod,O3}$ is largest in Category I (0.48) and smallest in Category II and IV (0.10) while $f_{mod,OH}$ is largest in Category IV (0.30), and smallest in Category I (0.04). Sulfate formation from in-cloud metal-catalyzed oxidation by $O_2$ and direct emission of anthropogenic sulfate contribute less than 1% of total sulfate in our samples and thus will be neglected in the discussion below.

Based on the modeled fractional contributions to the sulfate burden from each oxidant except HOX, and knowledge about reaction rate constants of "HOX + S(IV)" reactions and Henry's law constants of HOX, we calculate the amount of HOX needed to explain the discrepancy between $\Delta^{17}O_{mod}(nssSO_4^{2-})$ and $\Delta^{17}O_{obs}(nssSO_4^{2-})$ in Sect. 5.2.4.

## 5 Discussion

### 5.1 Observations of $\Delta^{17}O(nssSO_4^{2-})$ and $nssSO_4^{2-}$ concentration

Our observations of $nssSO_4^{2-}$ concentration (0.2-3.5 µg m$^{-3}$) are consistent with those (<2 µg m$^{-3}$) over the Southern Ocean measured by Sievering et al. (2004). Other published data for samples with air originating from the remote Atlantic Ocean showed a $[SO_4^{2-}]_{nss}$ between 0.9 and 4.5 µg m$^{-3}$ (Alexander et al., 2012), consistent with our observations. Higher observed $[SO_4^{2-}]_{nss}$ between 50°S - 70°S compared to 20°S - 40°S could be due to a higher DMS emission flux over 50°-70°S (Boucher et al., 2003).

Previous studies have suggested a large contribution to sulfate formation from $O_3$ oxidation in the MBL as the aqueous-phase reaction between S(IV) and $O_3$ is very fast at pH > 5 (Chameides and Stelson, 1992; Sievering et al, 1991; 2004; O'Dowd et al., 2000; Alexander et al., 2012). These studies did not consider the HOX mechanism due to the large uncertainty in the reaction rates. As the reaction of HOX with $SO_3^{2-}$ is also fast ($k_{HOBr+SO_3^{2-}} = 5\times10^9$ $M^{-1}s^{-1}$ and $k_{HOCl+SO_3^{2-}} = 7.6\times10^8$ $M^{-1}s^{-1}$), cloud pH > 5 will promote rapid aqueous-phase sulfate formation by HOX in addition to $O_3$. A large contribution from $O_3$ will yield a high $\Delta^{17}O(nssSO_4^{2-})$ value (6.5 ‰), but most samples in this study have low $\Delta^{17}O(nssSO_4^{2-})$ values (0.7±0.4 ‰). Thus, our results argue against a significant role of sulfate formation via $O_3$ oxidation in remote MBL. Indeed, our calculated $O_3$ contribution range ($f_{O3,min}=0.03, f_{O3,max}=0.12$) is more consistent with that reported by von Glasow et al. (2002), who did consider HOX. Their simulations of sulfate production in the MBL yielded a calculated $O_3$ contribution of 2-8% to the total sulfate production, while HOX contributed about 30%.

In comparison, Alexander et al. (2012) showed observations of $\Delta^{17}O(nssSO_4^{2-})$ of 1.1-1.4 ‰ for samples with back trajectories over the Iberian Peninsula during summer and 2.2-7.3 ‰ for samples with back trajectories over the Atlantic Ocean during winter. Their results suggested sulfate formation via HOX oxidation is not significant over subtropical northeast Atlantic during winter but potentially important in the more polluted coastal location of the Iberian Peninsula during summer. Our samples show lower $\Delta^{17}O(nssSO_4^{2-})$ than Alexander et al. (2012) in general, but were collected at different locations and during different seasons.

## 5.2 Comparison of modeled *versus* observed $\Delta^{17}O(nssSO_4^{2-})$

As shown in Fig 3a, the standard model significantly overestimates observations of $\Delta^{17}O(nssSO_4^{2-})$. This could be caused by the model (1) overestimating ($O_3$) or underestimating (OH, $H_2O_2$) oxidant abundances, (2) overestimating the amount of clouds, (3) overestimating the pH of clouds, or (4) neglecting sulfate formation from HOX oxidation. In this section, we examine each of these possibilities.

### 5.2.1 Oxidant sensitivity simulations

To investigate the impact of model biases in oxidant concentrations on calculated $\Delta^{17}O(nssSO_4^{2-})$, we perform three sensitivity runs by (1) doubling OH concentrations, (2) halving $O_3$ concentration and (3) doubling $H_2O_2$ concentration everywhere in the model. Figure 3b-d show $\Delta^{17}O_{mod}(nssSO_4^{2-})$ versus $\Delta^{17}O_{obs}(nssSO_4^{2-})$ for these three sensitivity runs. The discrepancy between $\Delta^{17}O_{mod}(nssSO_4^{2-})$ and $\Delta^{17}O_{obs}(nssSO_4^{2-})$ is not reconciled by changing the oxidant concentrations in the model. On average, $\Delta^{17}O_{mod}(nssSO_4^{2-})$ changes to 1.9±1.3 ‰, 1.8±1.2 ‰ and 1.4±0.9 ‰ for doubling OH concentration, halving $O_3$ concentration and doubling $H_2O_2$ concentration, respectively, compared to 1.8±1.2 ‰ for the standard run.

Doubling modeled OH concentrations results in an increase in the mass fraction of sulfate formed via gas-phase oxidation by OH ($f_{mod,OH}$) from 0.20 to 0.27 and a decrease via aqueous-phase oxidation by $H_2O_2$ and $O_3$ ($f_{mod,aq}=f_{mod,H2O2}+f_{mod,O3}$) from

0.77 to 0.70 (Table 3a). This would be expected to result in lower $\Delta^{17}O_{mod}(nssSO_4^{2-})$ as long as the relative importance of $H_2O_2$ and $O_3$ in the aqueous phase remains constant. However, doubling OH results in a small increase in $f_{mod,O3}$ from 0.20 to 0.22 (Table 3a). The small increase in $f_{mod,O3}$ occurs because of the reduction in the aqueous-phase sulfate production rate, which is caused by lower S(IV) due to faster removal of $SO_2$ by OH in the gas phase. A reduction in the aqueous-phase sulfate formation rate results in higher cloud-water pH, which increases the fraction of sulfate produced from $O_3$ oxidation, even though the total aqueous-phase sulfate production decreases. Thus, doubling OH concentrations has an insignificant effect on $\Delta^{17}O_{mod}(nssSO_4^{2-})$. The nighttime OH concentrations observed in forests and urban areas (Faloona et al., 2001; Lu et al., 2014) should also have insignificant effects on our model results as most of our samples are collected in the remote MBL where isoprene and VOCs abundances are low. Our sensitivity study with doubled OH suggests additional nighttime sources of OH would not resolve the modeled overestimate of $\Delta^{17}O(nssSO_4^{2-})$ observations.

Halving modeled $O_3$ concentrations results in a small decrease in $f_{mod,O3}$ from 0.20 to 0.19 and a change of less than 0.01 in $f_{mod,aq}$ (Table 3b). The decrease in $f_{mod,O3}$ is small because $f_{mod,O3}$ is mainly regulated by the cloud pH rather than $O_3$ abundance. In other words, the sulfate burden from $O_3$ oxidation is limited by concentration of $SO_3^{2-}$, not $O_3$. As a result, halving $O_3$ has an insignificant effect on $\Delta^{17}O_{mod}(nssSO_4^{2-})$.

Doubling modeled $H_2O_2$ concentrations results in an increase in $f_{mod,H2O2}$ from 0.57 to 0.66, a decrease in $f_{mod,O3}$ from 0.20 to 0.14 (Table 3c) and an increase in $f_{mod,aq}$ from 0.77 to 0.80. The increase in $f_{mod,H2O2}$ causes an increase in $\Delta^{17}O_{mod}(nssSO_4^{2-})$ of less than 0.1 ‰, which is a small effect compared to the change in $f_{mod,O3}$ that results in a decrease in $\Delta^{17}O_{mod}(nssSO_4^{2-})$ of 0.4 ‰. The decrease in $f_{mod,O3}$ is caused by the decrease in cloud pH due to higher aqueous-phase sulfate production rate. Although doubling $H_2O_2$ results in a decrease in $\Delta^{17}O_{mod}(nssSO_4^{2-})$ of 0.4 ‰ on average, it is still too small to reconcile the 1.1 ‰ discrepancy between $\Delta^{17}O_{mod}(nssSO_4^{2-})$ and $\Delta^{17}O_{obs}(nssSO_4^{2-})$.

### 5.2.2 Cloud fraction sensitivity simulations

To assess the uncertainty regarding the modeled cloud amount, we perform a sensitivity study by halving the cloud fraction in the model. As in-cloud S(IV) oxidation by $H_2O_2$ and $O_3$ produces sulfate with $\Delta^{17}O > 0$, a decrease in cloud fraction can potentially reduce $\Delta^{17}O_{mod}(nssSO_4^{2-})$ by reducing in-cloud sulfate formation.

Figure 3e shows the comparison between $\Delta^{17}O_{mod}(nssSO_4^{2-})$ and $\Delta^{17}O_{obs}(nssSO_4^{2-})$ for the sensitivity simulation where cloud fraction is halved. The discrepancy between the model and observations is similar to that in the standard run. Averaged $\Delta^{17}O_{mod}(nssSO_4^{2-})$ for samples in different categories are shown in Table 3d. Compared to the standard run, averaged $\Delta^{17}O_{mod}(nssSO_4^{2-})$ for all samples changes less than 0.1 ‰. Halving cloud fraction results in a decrease in $f_{mod,aq}$ from 0.77 to 0.70 and an increase in $f_{mod,OH}$ from 0.20 to 0.26. The change of $f_{mod,O3}$ is less than 0.01. A large decrease of $f_{mod,O3}$ is not observed by halving cloud fraction because lowering aqueous-phase sulfate production rates results in higher cloud pH,

shifting the relative importance of $H_2O_2$ and $O_3$ oxidation in the aqueous phase. Halving the cloud amount thus does not resolve the discrepancy between $\Delta^{17}O_{mod}(nssSO_4^{2-})$ and $\Delta^{17}O_{obs}(nssSO_4^{2-})$.

### 5.2.3 Cloud pH sensitivity simulations

Previous work has shown that bulk cloud models tend to underestimate sulfate formed via the $O_3$ pathway by underestimating pH and thus underestimating the fraction of S(IV) that is in the form of $SO_3^{2-}$. Yuen et al. (1996) developed a parameterization to correct for the underestimate in cloud pH by comparing a bulk cloud model with a cloud model that resolves the heterogeneity in cloud chemistry. The uncertainty in this parameterization, and thus the degree to which it might result in an overestimate of the contribution of $O_3$ to in-cloud sulfate formation in GEOS-Chem is difficult to assess.

We perform a sensitivity study which neglects heterogeneity in cloud chemistry by turning off the Yuen et al. (1996) parameterization. By using only bulk cloud pH calculations, this is effectively a lower limit for cloud pH, and thus is a lower limit for the contribution of $O_3$ to in-cloud sulfate formation in the model in the absence of HOX.

 Figure 3f shows the comparison between $\Delta^{17}O_{mod}(nssSO_4^{2-})$ and $\Delta^{17}O_{obs}(nssSO_4^{2-})$ for this low cloud pH simulation. Overall, $\Delta^{17}O_{mod}(nssSO_4^{2-})$ matches $\Delta^{17}O_{obs}(nssSO_4^{2-})$ much better than the standard run, especially for samples over

Southern Ocean. $\Delta^{17}O_{mod}(nssSO_4^{2-})$ ranges from 0.4 ‰ to 2.4 ‰, with an average of 1.1±0.5 (1σ) ‰. In comparison, $\Delta^{17}O_{obs}(nssSO_4^{2-})$ ranges from 0.0 ‰ to 1.6 ‰, with an average of 0.7±0.4 (1σ) ‰. The difference between averaged $\Delta^{17}O_{mod}(nssSO_4^{2-})$ and $\Delta^{17}O_{obs}(nssSO_4^{2-})$ (0.4 ‰) is just slightly larger than measurement uncertainty (± 0.3 ‰.). Compared to the standard run, $f_{mod,O3}$ (0.07 vs. 0.20) is much smaller. $f_{mod,O3}$ of 0.07 is within the $O_3$ contribution fraction range constrained by the observations ($f_{min,O3}$=0.03, $f_{max,O3}$=0.12). The decrease of $f_{mod,O3}$ is the main reason for the drop in

$\Delta^{17}O_{mod}(nssSO_4^{2-})$. For samples over Southern Ocean, $f_{mod,O3}$ decreases from 0.48 to 0.08 and $\Delta^{17}O_{mod}(nssSO_4^{2-})$ decreases from 3.6 ‰ to 1.3 ‰ correspondingly (Table 3e).

 Though averaged $\Delta^{17}O_{mod}(nssSO_4^{2-})$ is not much larger than $\Delta^{17}O_{obs}(nssSO_4^{2-})$ in the low cloud pH sensitivity study, the model does a poor job matching observations for samples with $\Delta^{17}O_{obs}(nssSO_4^{2-})$ < 0.7 ‰. 25 out of 31 samples with $\Delta^{17}O_{obs}(nssSO_4^{2-})$ smaller than 0.7 ‰ show that $\Delta^{17}O_{mod}(nssSO_4^{2-})$ overestimates $\Delta^{17}O_{obs}(nssSO_4^{2-})$ by more than 50%

(0.9 ‰), which indicates the model's inability to produce sulfate with low $\Delta^{17}O$ even while underestimating cloud pH. The majority of the discrepancy occurs for samples in the Southern Ocean (Catogory I) and subtropical MBL (Catogory III).

### 5.2.4 Contribution of HOX oxidation to sulfate formation

We can estimate the fractional contribution of HOX ($f_{obs,HOX}$) to total sulfate abundance necessary to explain the low $\Delta^{17}O_{obs}(nssSO_4^{2-})$ by comparing modeled $\Delta^{17}O(nssSO_4^{2-})$ with observations. $f_{obs,HOX}$ is calculated as shown below:

$$f_{obs,HOX} = 1 - \frac{\Delta^{17}O_{obs}(nssSO_4^{2-})}{\Delta^{17}O_{mod}(nssSO_4^{2-})} \qquad (8)$$

The derivation of Eq. (8) is described in the Appendix A. Calculating $f_{obs,HOX}$ using Eq. (8) may overestimate $f_{obs,HOX}$, as it assumes that the addition of "S(IV)+HOX" will not impact cloud pH. We estimate that this assumption overestimates calculation of $f_{obs,HOX}$ by about 15% (see Appendix A).

For $\Delta^{17}O_{mod}(nssSO_4^{2-})$ in Eq. (8), we use results from both the standard run in Sect. 4.2 and the low cloud pH run in Sect.
5.2.3 to place bounds on $f_{obs,HOX}$, using the low cloud pH sensitivity study as a lower limit for $f_{obs,HOX}$. We assume $f_{obs,HOX}=0$ when $\Delta^{17}O_{obs}(nssSO_4^{2-}) > \Delta^{17}O_{mod}(nssSO_4^{2-})$ (12 samples in the standard run and 22 samples in the low cloud pH run). $f_{obs,HOX}$ for each sample is shown in Fig. 4 for both runs. The averaged $f_{obs,HOX}$ is shown for samples in each category in Table 4a (standard run) and 4b (low cloud pH run). Among all samples, averaged $f_{obs,HOX}$ is 0.50±0.33 for the standard run and 0.33±0.32 for the low cloud pH run. $f_{obs,HOX}$ is largest for samples over the Southern Ocean (0.58-0.84 on average),
followed by the subtropical MBL (0.36-0.47 on average). $f_{obs,HOX}$ is lowest for samples collected near tropical coasts in the standard run (0.35) and near the Antarctic coast in the low cloud pH run (0.18).

We can estimate the concentration of HOX needed to achieve $f_{obs,HOX}$ using Eq. (9) below. The derivation of Eq. (9) is described in the Appendix A.

$$[HOX]_{aq} = \frac{f_{obs,HOX}}{\frac{k_{HOX+HSO_3^-}f_{obs,H2O2}}{k_{H2O2+HSO_3^-}[H_2O_2]_{aq}} + \frac{k_{HOX+SO_3^{2-}}f_{obs,O3}}{k_{O3+SO_3^{2-}}[O_3]_{aq}}} \tag{9}$$

where $k_{H2O2+HSO_3^-} = 2.4\times10^3$ M$^{-1}$s$^{-1}$ at pH = 4.5 and $k_{O3+SO_3^{2-}} = 1.5\times10^9$ M$^{-1}$s$^{-1}$. $k_{HOBr+SO_3^{2-}} = 5\times10^9$ M$^{-1}$s$^{-1}$ and $k_{HOCl+SO_3^{2-}} = 7.6\times10^8$ M$^{-1}$s$^{-1}$ are from Fogelman et al. (1989) and Troy and Margerum (1991), respectively. We assume $k_{HOX+SO_3^{2-}} = 2.9\times10^9$ M$^{-1}$s$^{-1}$ as the average of $k_{HOBr+SO_3^{2-}}$ (=$5\times10^9$ M$^{-1}$s$^{-1}$, Troy and Margerum (1991)) and $k_{HOCl+SO_3^{2-}}$(= $7.6\times10^8$ M$^{-1}$s$^{-1}$, Fogelman et al. (1989)). We assume $k_{HOX+HSO_3^-} = 2.0\times10^9$M$^{-1}$s$^{-1}$ which is the average of $k_{HOBr+HSO_3^-}$ (= $3.2\times10^9$ M$^{-1}$s$^{-1}$, upper limit from Liu (2000)) and $k_{HOCl+HSO_3^-}$ (=$k_{HOCl+SO_3^{2-}}$, as assumed by Vogt et al.
(1996) and von Glasow et al. (2002)). $f_{obs,HOX}$ is calculated from Eq. (8), and $f_{obs,H2O2}$ and $f_{obs,O3}$ are calculated from Eqs. (A9 and A14) using the same assumption as $f_{obs,HOX}$ (see Appendix A). $[H_2O_2]_{aq}$ and $[O_3]_{aq}$ are obtained from the model, and range from 2-172 μM and 113-463 pM, respectively. The range and median value of $[HOX]_{aq}$ for samples in different categories are shown in Table 4. $[HOX]_{aq}$ needed to explain $f_{obs,HOX}$ is on the order of 100 pM and 10 pM for the standard run and low cloud pH run, respectively. By assuming the Henry's law constant $H_{HOX}$ of 975 M atm$^{-1}$ (average between $H_{HOCl}$ and
$H_{HOBr}$ from Huthwelker et al. (1995) and Sander et al. (2006)), the daily-averaged gas-phase [HOX] mixing ratio $[HOX]_g$ is calculated and shown in Table 4. Due to the low solubility of HOX, under typical atmospheric conditions, more than 99% of total HOX is in the gas phase. Daily-averaged $[HOX]_g$ is on the order of 0.1 ppt and 0.01 ppt when using $f_{obs,HOX}$ from the standard run and low cloud pH run, respectively. Thus, a gas-phase HOX mixing ratio of $\approx 0.1$ ppt or higher would be sufficient to explain the observed $\Delta^{17}O(nssSO_4^{2-})$ of our samples. Uncertainties in our calculated $[HOX]_g$ can originate from

(1) uncertainties in the rate constant for reaction between HOX and $HSO_3^-$ ($k_{HOX+HSO_3^-}$), (2) uncertainties in the Henry's law constant for HOX ($H_{HOX}$) and (3) the efficiency of reactive uptake of gas-phase HOX onto cloud droplets that is not accounted for in our assumption of equilibrium of HOX between the gas and aqueous phase.

    In comparison, a box-modeling study by Vogt et al. (1996) estimated that daytime-averaged $[HOX]_g$ on the order of 10 ppt is needed to achieve a similar fraction ($\approx 60\%$) of sulfate formed via HOX oxidation. The difference in $[HOX]_g$ between our

study and Vogt et al. (1996) is caused by several factors. First, $H_{HOBr}$ in our calculations is an order of magnitude larger than that in Vogt et al. (1996), so that our calculations require an order of magnitude lower $[HOBr]_g$ to produce the same aqueous-phase concentration. Second, $H_2O_2$ and $O_3$ mixing ratios in our calculations (Eq. 9) are lower than those in Vogt et al. (1996) (0.6 ppt vs. 0.8 ppt for $[H_2O_2]$ and 18 ppt vs. 40 ppt for $[O_3]$), so that higher $[HOBr]_g$ is needed in Vogt et al. (1996) to compete with S(IV) oxidation by $H_2O_2$ and $O_3$. Third, in our simple calculation we assume equilibrium of HOX between

the gas- and aqueous-phase while Vogt et al. (1996) considers all mass transfer limitations. Higher $[HOX]_g$ will be calculated if diffusion and subsequent mass accommodation of gas-phase HOX onto the cloud droplets is not fast enough to compensate for the loss of HOX from aqueous-phase chemistry. Fourth, $[HOX]_g$ on the order of 0.1 ppt calculated in our study is a daily-averaged concentration while $[HOX]_g$ on the order of 10 ppt in Vogt et al. (1996) is a daytime-averaged concentration, and hence are not directly comparable. Vogt et al. (1996) has shown that nighttime-averaged $[HOX]_g$ is about

two orders of magnitude lower than daytime-averaged $[HOX]_g$.

    The daily-averaged HOBr mixing ratio over Southern Ocean (40°~64°S, below 100 m) modeled by Schmidt et al. (2016) varied from 0.1-0.3 ppt to 0.2-0.4 ppt for simulations without and with debromination from acidic sea salt aerosol, respectively. The HOX mixing ratio on the order of 0.1 ppt calculated from $\Delta^{17}O(nssSO_4^{2-})$ of our samples using standard run results is consistent with that obtained in Schmidt et al. (2016).

Comparison of our calculated daily-averaged HOX mixing ratios with observations is difficult, as HOX is expected to show significant diurnal variability (on the order of 2 ppt), with mixing ratios near zero at night and peaking at around noon (von Glasow et al., 2002). A daytime-averaged HOBr mixing ratio of about 10 ppt was observed by Liao et al. (2012) at Alaska in spring, which is about 2~3 orders of magnitude higher than our calculated daily-averaged $[HOX]_g$. The nighttime HOBr mixing ratio in their study was below the detection limit of about 2 ppt. A more recent study that measured HOBr

mixing ratios along the flight tracks over the tropical West Pacific during January and February 2014 showed a daily-averaged HOBr mixing ratio of about 2 ppt, with a lower detection limit of about 0.2 ppt (Le Breton et al., 2016). This is still much higher than our calculated daily-averaged $[HOX]_g$, but it is likely that HOBr mixing ratios could vary significantly with sampling locations and sampling time (Schmidt et al., 2016). Field campaigns of HOX measurements are necessary to assess our calculated HOX mixing ratios over our sampling regions.

## 6 Conclusion

This study uses a combination of observations and modeling of $\Delta^{17}O(nssSO_4^{2-})$ to quantify the role of HOX (= HOBr + HOCl) in sulfate formation in the remote MBL. Samples collected over a wide spatial range in the MBL during spring and summer show low $\Delta^{17}O_{obs}(nssSO_4^{2-})$ (0.7±0.4 ‰), which suggests only 3%~12% of sulfate is formed via $O_3$ oxidation. The standard model run overestimates $\Delta^{17}O_{obs}(nssSO_4^{2-})$ by about a factor of 2.5 on average because it overestimates the amount of sulfate formed by $O_3$ in the aqueous phase. This discrepancy could not be resolved by either varying oxidant concentrations, halving cloud amount or using a lower limit for cloud pH in the model. Our calculations suggest that the discrepancy can be explained with a fractional contribution of sulfate abundance formed by HOX ranging from 33~50% over the entire area sampled, with the highest fraction (58~84%) in the Southern Ocean MBL. A daily-averaged gas-phase HOX mixing ratio of $\approx 0.1$ ppt or higher would be sufficient to explain the observed $\Delta^{17}O(nssSO_4^{2-})$ of our samples. This study provides the first observational constraint on the role of hypohalous acids in sulfate aerosol formation in the MBL. Future studies will implement the "S(IV) + HOX" reaction into GEOS-Chem to investigate the impacts of this reaction on the global sulfur budget and possible feedbacks on acid-catalyzed reactive halogen production.

### Acknowledgement

We thank Roland von Glasow and Jon Abbatt for helpful discussions in the planning phase of this project. This project is funded by NSF AGS 1343077. The sampling during the Malaspina cruise was funded by the Spanish Ministry of Economy and Competiveness (Circumnavigation Expedition Malaspina 2010: Global Change and Biodiversity Exploration of the Global Ocean. CSD2008-00077). Z.Q. Xie acknowledges the support from the NSFC 91544103. Q. Chen thanks Viral Shah for help on GEOS-Chem modeling.

## Appendix A

Hitherto, there is no observational constraint on HOX mixing ratios in the mid- and high-latitude remote MBL. Models have suggested a large range of HOX mixing ratios on the order of 0.1 ppt (Schmidt et al., 2016) to 10 ppt (Vogt et al., 1996) over these regions. Here we quantify the daily-averaged HOX mixing ratio indirectly from observed and modeled $\Delta^{17}O$ of sulfate. First, we calculate the fractional contribution of the HOX oxidation pathway ($f_{obs,HOX}$) to sulfate abundance in our samples. Then, we calculate the HOX mixing ratio needed to achieve this $f_{obs,HOX}$.

### A1 Calculation of $f_{obs,HOX}$

We assume all modeled sulfate in the MBL is formed via gas-phase OH oxidation and aqueous-phase $H_2O_2$ or $O_3$ oxidation based on the insignificant contribution (<3%) of other sulfate sources in the model.

$$f_{mod,OH} + f_{mod,H2O2} + f_{mod,O3} = 1 \tag{A1}$$

$$af_{mod,H2O2} + bf_{mod,O3} = \Delta^{17}O_{mod}(nssSO_4^{2-}) \tag{A2}$$

where $a = \Delta^{17}O(nssSO_4^{2-})_{H2O2}= 0.7\ ‰$ and $b = \Delta^{17}O(nssSO_4^{2-})_{O3} = 6.5\ ‰$. For the observations, we assume all sulfate in the MBL is formed via gas-phase OH oxidation, aqueous-phase $H_2O_2$, $O_3$ and HOX oxidation pathways:

$$f_{obs,OH} + f_{obs,H2O2} + f_{obs,O3} + f_{obs,HOX} = 1 \tag{A3}$$

$$af_{obs,H2O2} + bf_{obs,O3} = \Delta^{17}O_{obs}(nssSO_4^{2-}) \tag{A4}$$

where $f_{obs,OH}, f_{obs,H2O2}, f_{obs,O3}$ and $f_{obs,HOX}$ are the observed fractions of sulfate formed via gas-phase OH, aqueous-phase $H_2O_2$, $O_3$ and HOX oxidation pathways, respectively. To solve for $f_{obs,HOX}$, two more equations involving $f_{obs,OH}, f_{obs,H2O2}, f_{obs,O3}$ and $f_{obs,HOX}$ are needed, in addition to Eqs. (A3-A4). Here we assume $f_{obs,O3}/f_{obs,H2O2}$ ratio is offset from $f_{mod,O3}/f_{mod,H2O2}$ ratio by $\Delta r_1$ and $f_{obs,OH}/f_{obs,H2O2}$ ratio is offset from $f_{mod,OH}/f_{mod,H2O2}$ ratio by $\Delta r_2$:

$$\frac{f_{obs,O3}}{f_{obs,H2O2}} = \frac{f_{mod,O3}}{f_{mod,H2O2}} + \Delta r_1 \tag{A5}$$

$$\frac{f_{obs,OH}}{f_{obs,H2O2}} = \frac{f_{mod,OH}}{f_{mod,H2O2}} + \Delta r_2 \tag{A6}$$

Combining Eqs. (A3-A6) and using Eqs. (A1-A2) yield:

$$f_{obs,HOX} = 1 - \frac{\Delta^{17}O_{obs}(nssSO_4^{2-})}{\Delta^{17}O_{mod}(nssSO_4^{2-})} + \Delta f \tag{A7}$$

where

$$\Delta f = f_{obs,H2O2}[(\frac{b}{\Delta^{17}O_{mod}(nssSO_4^{2-})} - 1)\Delta r_1 - \Delta r_2] \tag{A8}$$

and

$$f_{obs,H2O2} = \frac{f_{mod,H2O2}\Delta^{17}O_{obs}(nssSO_4^{2-})}{\Delta^{17}O_{mod}(nssSO_4^{2-})+b\Delta r_1 f_{mod,H2O2}} \tag{A9}$$

Setting $\Delta f=0$ yields Eq. (8) in Sect. 5.2.4. $\Delta f$ is zero when $\Delta r_1= \Delta r_2=0$, which effectively assumes that the decreases in $f_{mod,OH}$, $f_{mod,H2O2}$ and $f_{mod,O3}$ after adding HOX in the model could be proportional to their relative fractions in the model. $\Delta r_1$ will be zero if cloud pH is unchanged, i.e. the S(IV) partitioning will remain unchanged after adding "S(IV) + HOX" reaction. The potential magnitude of $\Delta f$, which is dependent on the relative magnitude of $\Delta r_1$ and $\Delta r_2$, is discussed below.

$\Delta r_1$ is expected to be negative with the addition of "S(IV) + HOX" reaction. Additional sulfate production in the aqueous phase will decrease cloud pH, resulting in decreases in the fractional contribution of $O_3$ relative to $H_2O_2$ ($f_{O3}/f_{H2O2}$). The

magnitude of the potential decrease in cloud pH can only be obtained after adding the "S(IV) + HOX" reactions in the model, which will be done in a follow-up study.

$\Delta r_2$ is expected to be positive with the addition of "S(IV) + HOX" reaction. HOX competes with $H_2O_2$ during oxidation of $HSO_3^-$ in clouds, which causes a direct decrease in the fraction of sulfate formed via $H_2O_2$ oxidation ($f_{mod,H2O2}$). On the other hand, gas-phase sulfate production from oxidation of $SO_2$ by OH occurs mainly in the absence of clouds. Thus, adding "S(IV) + HOX" reaction causes an indirect decrease in the fraction of sulfate formed via OH oxidation ($f_{mod,OH}$) by increasing in-cloud and consequent total sulfate production, which depends on the availability of S(IV). Our model indicates that in-cloud

sulfate production is limited by S(IV) abundance among our sampling locations (see the doubling $H_2O_2$ and $O_3$ simulation below), such that the decrease of $f_{mod,OH}$ is small compared to that of $f_{mod,H2O2}$ with the addition of "S(IV) + HOX" reaction, which results in an increase in the fractional contribution of OH relative to $H_2O_2$ ($f_{OH}/f_{H2O2}$).

     As $\Delta^{17}O_{mod}(nssSO_4^{2-}) < b$, the term $(\frac{b}{\Delta^{17}O_{mod}(nssSO_4^{2-})} - 1)$ is positive. In addition, $f_{obs,H2O2}$ is positive. Thus, $\Delta f$ is likely a negative number, which indicates Eq. (8) in Sect. 5.2.4 may overestimate $f_{obs,HOX}$.

We simulate the effect of an additional aqueous-phase reaction ("S(IV) + HOX") in the model on $\Delta r_1$ and $\Delta r_2$ by doubling both $H_2O_2$ concentration and $O_3$ concentration. By doing this, we attribute half of the sulfate produced via $H_2O_2$ and $O_3$ oxidation to HOX oxidation. This simulation yields fractional contribution of sulfate formed via gas-phase OH, aqueous-phase $H_2O_2$, $O_3$ and HOX oxidation pathways to be 0.17, 0.33, 0.08 and 0.40, respectively, among all sampling locations. Compared to the standard run, $f_{mod,O3}/f_{mod,H2O2}$ ratio decreases from 0.35 to 0.23 and $f_{mod,OH}/f_{mod,H2O2}$ ratio increases from 0.35

to 0.53 on average among all sampling locations, yielding $\Delta r_1$=-0.12 and $\Delta r_2$=0.18. Substituting standard run results ($\Delta^{17}O_{obs}(nssSO_4^{2-})$=0.7 ‰, $\Delta^{17}O_{mod}(nssSO_4^{2-})$=1.8 ‰ and $f_{mod,H2O2}$=0.57 (Table 2)) and $\Delta r_1$=-0.12 and $\Delta r_2$=0.18 into Eq. (A8-A9) yields $\Delta f$=-0.15. Thus, if the addition of "S(IV) + HOX" reaction results in 40% of sulfate formed via oxidation of S(IV) by HOX, then our estimate of $f_{obs,HOX}$ using Eq. (8) in Sect. 5.4.2 would be 15% too high. The fraction (40%) of sulfate formed via oxidation of S(IV) by HOX is within the range of the averaged $f_{obs,HOX}$ (33-50%) calculated from observed and

modeled $\Delta^{17}O(nssSO_4^{2-})$ using Eq. (8). Thus, we suggest that calculating $f_{obs,HOX}$ for our samples using Eq. (8) may overestimate $f_{obs,HOX}$ by about 15%. This is smaller than the difference in $f_{obs,HOX}$ calculated from our low cloud pH (0.33±0.32) and standard model (0.50±0.33) runs. The actual magnitude of $\Delta f$ can only be obtained by implementing the "S(IV)+HOX" reaction in the model.

**A2 Calculation of HOX mixing ratios**

We estimate the mixing ratio of HOX needed to achieve $f_{obs,HOX}$. First, we divide $f_{obs,HOX}$ into two parts:

$$f_{obs,HOX} = f_{1,HOX} + f_{2,HOX} \qquad\qquad (A10)$$

where $f_{1,\text{HOX}}$ and $f_{2,\text{HOX}}$ are fractional contributions from "HOX + $HSO_3^-$" reaction and "HOX + $SO_3^{2-}$" reaction, respectively. $H_2O_2$ reacts with $HSO_3^-$ only and $O_3$ reacts mainly with $SO_3^{2-}$ (Hoffmann and Calvert, 1985), while HOX reacts quickly with both $HSO_3^-$ and $SO_3^{2-}$. Then we compare $f_{1,\text{HOX}}$ with $f_{\text{obs,H2O2}}$ and $f_{2,\text{HOX}}$ with $f_{\text{obs,O3}}$:

$$\frac{f_{1,\text{HOX}}}{f_{\text{obs,H2O2}}} = \frac{k_{\text{HOX}+HSO_3^-}[\text{HOX}]_{\text{aq}}}{k_{\text{H2O2}+HSO_3^-}[\text{H}_2\text{O}_2]_{\text{aq}}} \tag{A11}$$

$$\frac{f_{2,\text{HOX}}}{f_{\text{obs,O3}}} = \frac{k_{\text{HOX}+SO_3^{2-}}[\text{HOX}]_{\text{aq}}}{k_{\text{O3}+SO_3^{2-}}[\text{O}_3]_{\text{aq}}} \tag{A12}$$

where $k_{\text{H2O2}+HSO_3^-}$ and $k_{\text{HOX}+HSO_3^-}$ are rate coefficients for reactions of $H_2O_2$ and HOX with $HSO_3^-$, respectively; $k_{\text{O3}+SO_3^{2-}}$ and $k_{\text{HOX}+SO_3^{2-}}$ are rate coefficients for reactions of $O_3$ and HOX with $SO_3^{2-}$, respectively; and $[\text{H}_2\text{O}_2]_{\text{aq}}$, $[\text{O}_3]_{\text{aq}}$ and $[\text{HOX}]_{\text{aq}}$ are the aqueous-phase concentration of $H_2O_2$, $O_3$ and HOX in the cloud droplets, respectively. Combining Eqs. (A10-A12) yields:

$$[\text{HOX}]_{\text{aq}} = \frac{f_{\text{obs,HOX}}}{\frac{k_{\text{HOX}+HSO_3^-} f_{\text{obs,H2O2}}}{k_{\text{H2O2}+HSO_3^-}[\text{H}_2\text{O}_2]_{\text{aq}}} + \frac{k_{\text{HOX}+SO_3^{2-}} f_{\text{obs,O3}}}{k_{\text{O3}+SO_3^{2-}}[\text{O}_3]_{\text{aq}}}} \tag{A13}$$

This is the same equation as Eq. (9) in Sect. 5.4.2. $f_{\text{obs,HOX}}$ and $f_{\text{obs,H2O2}}$ are calculated in Eq. (A7) and Eq. (A9), respectively. Combining Eqs. (A5 and A9) yields:

$$f_{\text{obs,O3}} = \frac{(f_{\text{mod,O3}} + \Delta r_1 f_{\text{mod,H2O2}}) \Delta^{17}O_{\text{obs}}(\text{nssSO}_4^{2-})}{\Delta^{17}O_{\text{mod}}(\text{nssSO}_4^{2-}) + 6.5 \Delta r_1 f_{\text{mod,H2O2}}} \tag{A14}$$

For the calculations of $f_{\text{obs,HOX}}$, $f_{\text{obs,H2O2}}$ and $f_{\text{obs,O3}}$ in Sect. 5.4.2, both $\Delta r_1$ and $\Delta r_2$ are assumed to be zero.

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

**Tables**

Table 1. $\Delta^{17}O$ of sulfate produced via different pathways.

| Sulfate formation pathway | $\Delta^{17}O(nssSO_4^{2-})$ (‰) | References |
|---|---|---|
| $SO_2(g)+OH$ | 0 | Dubey et al., 1997; Lyons, 2001 |
| $S(IV)+H_2O_2$ | 0.7 | Savarino and Thiemens, 1999 |
| $S(IV)+O_3$ | 6.5 | Vicars and Savarino, 2014 |
| $S(IV)+O_2$ | -0.09 | Barkan and Luz, 2005 |
| $S(IV)+HOX$ | 0 | Fogelman et al., 1989; Troy and Margerum, 1991 |

Table 2. $[SO_4^{2-}]_{nss}$, $\Delta^{17}O_{obs}(nssSO_4^{2-})$, $f_{O3,min}$ and $f_{O3,max}$ obtained from observations and $\Delta^{17}O_{mod}(nssSO_4^{2-})$, $f_{mod,O3}$, $f_{mod,H2O2}$ and $f_{mod,OH}$ obtained from the standard model run for samples in different categories.

| Category | Characteristic | Number | $[SO_4^{2-}]_{nss}$ (µg m$^{-3}$) | $\Delta^{17}O_{obs}$ (‰) | $f_{O3,min}$ | $f_{O3,max}$ | $\Delta^{17}O_{mod}$ (‰) | $f_{mod,OH}$ | $f_{mod,H2O2}$ | $f_{mod,O3}$ | $f_{mod,het}$ |
|---|---|---|---|---|---|---|---|---|---|---|---|
| I | Southern Ocean | 13 | 1.4±0.8 | 0.5±0.3 | 0.01±0.02 | 0.08±0.05 | 3.6±1.1 | 0.04±0.04 | 0.46±0.16 | 0.48±0.19 | 0.02±0.02 |
| II | Antarctic coast | 18 | 1.4±0.7 | 0.7±0.4 | 0.03±0.04 | 0.11±0.06 | 1.1±0.3 | 0.25±0.11 | 0.65±0.09 | 0.10±0.04 | 0.01±0.01 |
| III | Subtropical MBL | 19 | 0.9±0.5 | 0.8±0.4 | 0.04±0.05 | 0.13±0.07 | 1.9±1.0 | 0.17±0.08 | 0.61±0.14 | 0.19±0.15 | 0.03±0.03 |
| IV | Tropical coasts | 17 | 1.3±0.9 | 0.8±0.4 | 0.03±0.04 | 0.13±0.06 | 1.2±0.4 | 0.30±0.16 | 0.54±0.15 | 0.10±0.06 | 0.03±0.03 |
| Total | / | 67 | 1.2±0.8 | 0.7±0.4 | 0.03±0.04 | 0.12±0.06 | 1.8±1.2 | 0.20±0.14 | 0.57±0.15 | 0.20±0.19 | 0.02±0.03 |

**Table 3.** $\Delta^{17}O_{mod}(nssSO_4^{2-})$, $f_{mod,O3}$, $f_{mod,H2O2}$ and $f_{mod,OH}$ obtained from five sensitivity studies: (a) double OH concentration, (b) halve $O_3$ concentration, (c) double $H_2O_2$ concentration, (d) halve clouds and (e) low cloud pH.

| Category | $\Delta^{17}O_{mod}$ (‰) | $f_{mod,OH}$ | $f_{mod,H2O2}$ | $f_{mod,O3}$ | $f_{mod,het}$ |
|---|---|---|---|---|---|
| (a) 2[OH] | | | | | |
| I | 3.8±1.1 | 0.06±0.05 | 0.39±0.14 | 0.53±0.18 | 0.02±0.02 |
| II | 1.3±0.4 | 0.29±0.11 | 0.57±0.09 | 0.13±0.05 | 0.01±0.01 |
| III | 1.8±1.0 | 0.25±0.11 | 0.51±0.13 | 0.20±0.16 | 0.03±0.03 |
| IV | 1.1±0.5 | 0.42±0.19 | 0.43±0.17 | 0.09±0.06 | 0.02±0.03 |
| Total | 1.9±1.3 | 0.27±0.17 | 0.48±0.15 | 0.22±0.20 | 0.02±0.02 |
| (b) 1/2[O₃] | | | | | |
| I | 3.5±1.2 | 0.04±0.04 | 0.46±0.17 | 0.47±0.20 | 0.02±0.02 |
| II | 1.2±0.3 | 0.25±0.11 | 0.65±0.09 | 0.10±0.04 | 0.01±0.01 |
| III | 1.7±1.0 | 0.17±0.08 | 0.63±0.14 | 0.17±0.16 | 0.03±0.03 |
| IV | 1.1±0.4 | 0.31±0.16 | 0.56±0.16 | 0.08±0.05 | 0.03±0.03 |
| Total | 1.8±1.2 | 0.20±0.14 | 0.58±0.15 | 0.19±0.19 | 0.02±0.03 |
| (c) 2[H₂O₂] | | | | | |
| I | 2.7±1.0 | 0.03±0.03 | 0.60±0.16 | 0.35±0.18 | 0.02±0.02 |
| II | 0.9±0.2 | 0.21±0.10 | 0.72±0.09 | 0.06±0.03 | 0.01±0.01 |
| III | 1.4±0.8 | 0.15±0.07 | 0.70±0.13 | 0.12±0.14 | 0.03±0.03 |
| IV | 1.0±0.3 | 0.27±0.15 | 0.61±0.16 | 0.07±0.03 | 0.03±0.03 |
| Total | 1.4±0.9 | 0.17±0.13 | 0.66±0.14 | 0.14±0.15 | 0.02±0.03 |
| (d) 1/2 clouds | | | | | |
| I | 3.7±1.1 | 0.06±0.05 | 0.42±0.14 | 0.50±0.18 | 0.02±0.03 |
| II | 1.1±0.3 | 0.31±0.12 | 0.58±0.11 | 0.10±0.04 | 0.01±0.01 |
| III | 1.9±1.0 | 0.23±0.10 | 0.53±0.13 | 0.19±0.15 | 0.04±0.04 |
| IV | 1.1±0.5 | 0.39±0.18 | 0.45±0.16 | 0.09±0.07 | 0.03±0.03 |
| Total | 1.8±1.2 | 0.26±0.17 | 0.50±0.15 | 0.20±0.19 | 0.03±0.03 |
| (e) Low cloud pH | | | | | |
| I | 1.3±0.4 | 0.07±0.03 | 0.81±0.09 | 0.08±0.06 | 0.03±0.04 |
| II | 0.6±0.1 | 0.26±0.10 | 0.72±0.10 | 0.01±0.00 | 0.01±0.01 |
| III | 1.4±0.5 | 0.18±0.07 | 0.68±0.10 | 0.10±0.05 | 0.03±0.04 |
| IV | 1.1±0.4 | 0.30±0.16 | 0.57±0.17 | 0.08±0.06 | 0.03±0.03 |
| Total | 1.1±0.5 | 0.21±0.13 | 0.69±0.14 | 0.07±0.06 | 0.02±0.03 |

**Table 4.** $f_{\text{obs,HOX}}$, $[\text{HOX}]_{\text{aq}}$ and $[\text{HOX}]_{\text{g}}$ calculated using the model results from (a) the standard run and (b) low cloud pH run, respectively. The mean and standard deviation are shown for $f_{\text{obs,HOX}}$. The range and median value are shown for $[\text{HOX}]_{\text{aq}}$ and $[\text{HOX}]_{\text{g}}$.

| Category | (a) standard run | | | (b) low cloud pH | | |
|---|---|---|---|---|---|---|
| | $f_{\text{obs,HOX}}$ | $[\text{HOX}]_{\text{aq}}$ (pM) | $[\text{HOX}]_{\text{g}}$ (ppt) | $f_{\text{obs,HOX}}$ | $[\text{HOX}]_{\text{aq}}$ (pM) | $[\text{HOX}]_{\text{g}}$ (ppt) |
| I | 0.84±0.15 | 41~30192 (102) | 0.04~30.97 (0.10) | 0.58±0.30 | 1~4048 (27) | 0.00~4.15 (0.03) |
| II | 0.42±0.29 | 0~395 (26) | 0.00~0.40 (0.03) | 0.18±0.26 | 0~145 (0) | 0.00~0.15 (0.00) |
| III | 0.47±0.34 | 0~3334 (97) | 0.00~3.42 (0.10) | 0.36±0.32 | 0~2585 (40) | 0.00~2.65 (0.04) |
| IV | 0.35±0.29 | 0~761 (85) | 0.00~0.78 (0.09) | 0.28±0.31 | 0~764 (54) | 0.00~0.78 (0.06) |
| Total | 0.50±0.33 | 0~30192 (88) | 0.00~30.97 (0.09) | 0.33±0.32 | 0~4048 (11) | 0.00~4.15 (0.01) |

# Figures

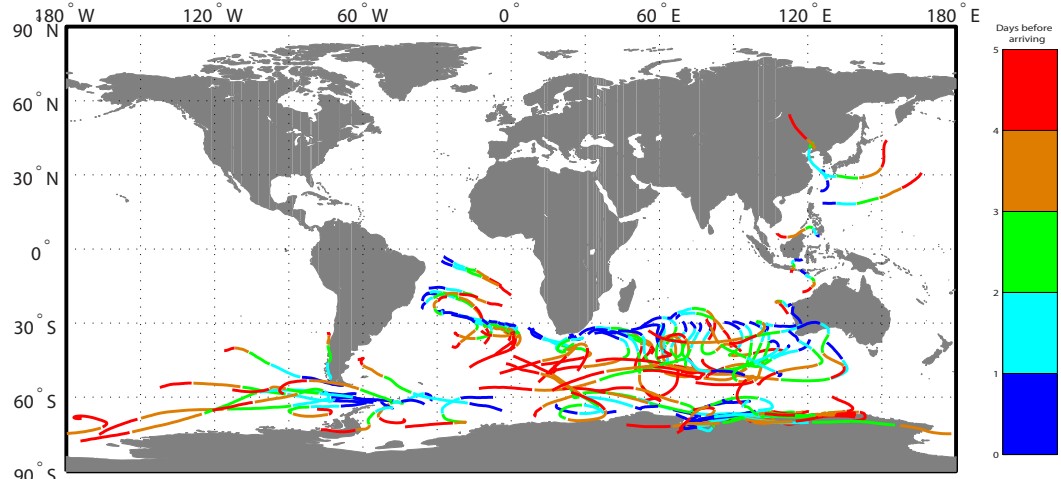

**Figure 1.** 5-day back trajectories calculated from the HYSPLIT model for all sampling locations. Blue indicates the ending point of each back trajectory.

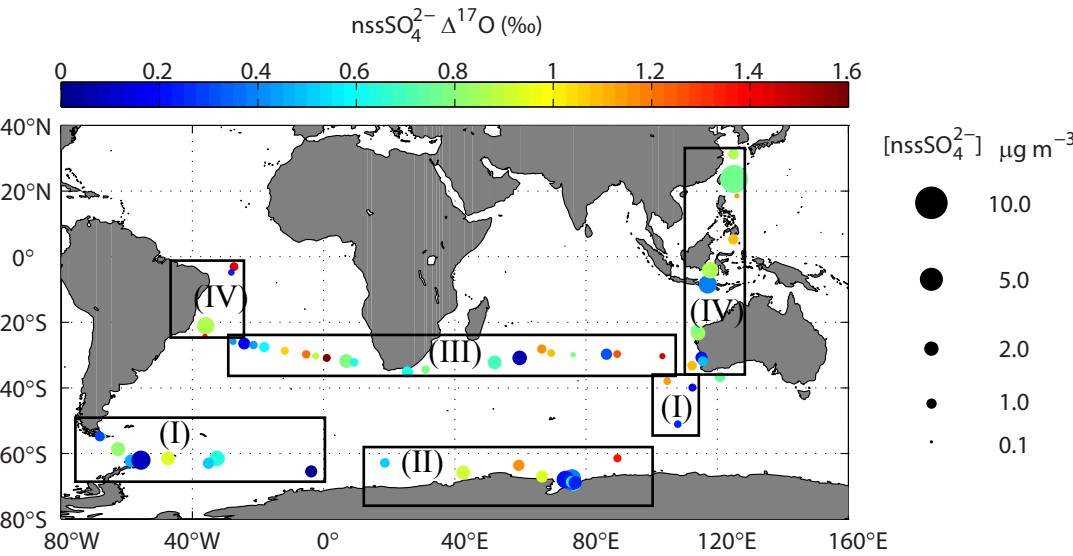

**Figure 2.** Observations of $\Delta^{17}O(nssSO_4^{2-})$ (‰) and $nssSO_4^{2-}$ concentration (µg m$^{-3}$) for aerosol samples collected in the MBL during spring and summer in 2011 and 2012. Black rectangles indicate regions I – IV.

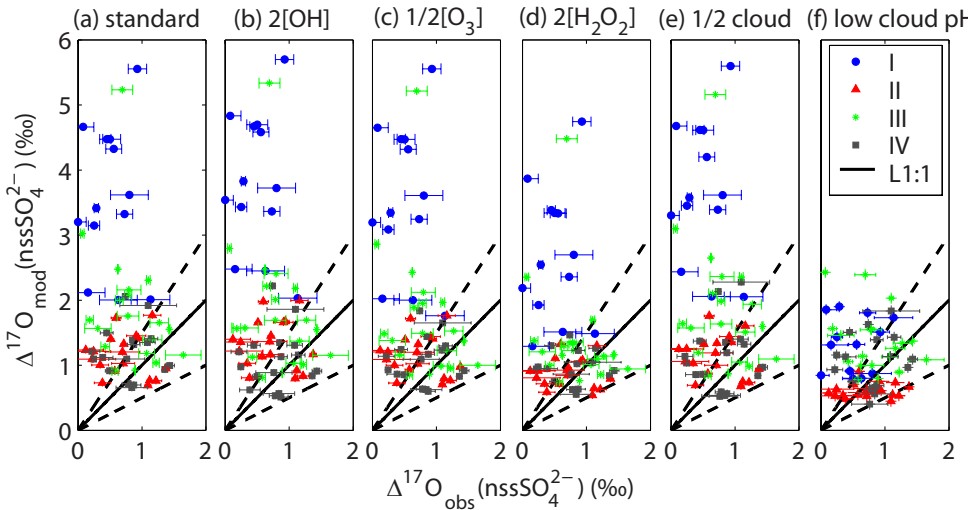

**Figure 3.** $\Delta^{17}O_{mod}(nssSO_4^{2-})$ versus $\Delta^{17}O_{obs}(nssSO_4^{2-})$ for different model simulations: (a) standard run, (b) double OH concentration, (c) halve $O_3$ concentration, (d) double $H_2O_2$ concentration, (e) halve clouds and (f) low cloud pH.

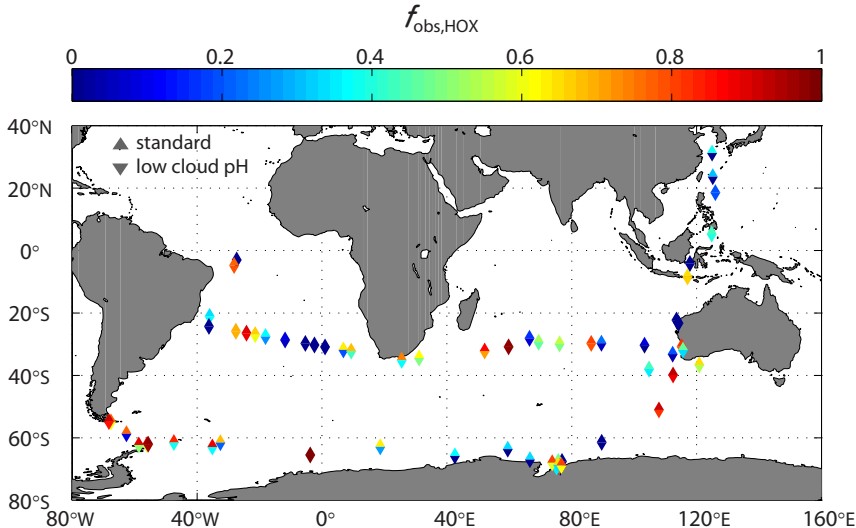

**Figure 4.** Calculated $f_{obs,HOX}$ for each sample using modeling results from the standard run (upward-pointing triangle) and low cloud pH run (downward-pointing triangle), respectively.