# Peer review of "Isotopic constraints on the role of hypohalous acids in sulfate aerosol formation in the remote marine boundary layer"

_Atmospheric Chemistry and Physics, 2016_

## Referee Comment (RC1) · R. Sander (Referee) · 12 Jul 2016

Chen et al. investigate the role of hypohalous acids on sulfate formation in the mbl. The study is very interesting and it should eventually be published in ACP if the following major issues can be resolved:

**Major Issues**

- I find it confusing that the manuscript switches between cloud droplets and aerosol particles. It looks like measurements of aerosols are compared to model
results of cloud chemistry. Is this the case? If yes, please explain why this approach can be valid.

- In Table 1, you cite Fogelman et al. (1989) and Troy and Margerum (1991) as references for $\Delta^{17}O = 0$ for the reactions S(IV) + HOX. However, I do not see any isotope chemistry in these papers. How do you derive this zero value? How would your results change if $\Delta^{17}O$ is non-zero for S(IV) + HOX?

- You conclude that a fraction of 33 to 50 % of the sulfate can be produced by about 0.1 pmol/mol HOX. Vogt et al. (1996), however, needed 100 times more HOX (10 pmol/mol) to achieve a similar fraction (60 %). Why is there such a big difference between these two studies? Can the difference be explained by different rate coefficients that were used? Were different reactions included in these studies? If not, what else could it be?

**Specific comments**

- Abstract: I think it should be mentioned here that uncertainties in reactive halogen concentrations are probably the main reason why halogen chemistry is excluded in large scale models (apart from uncertainties in the reaction rates).

- page 5, line 131: Here you list several ions which were measured. Unfortunately, the list does not include bromide. Since the reaction HOBr + S(IV) produces bromide, it would be very interesting to know bromide aerosol concentrations. Would it be possible to analyze the samples for bromide?

- page 6, line 173: Metal-catalyzed oxidation of S(IV) is mentioned here. Could you add the information which metals (Fe?, Mn?) are included in the model and how the model calculates their concentrations? I think this information is necessary to

understand why this pathway contributes less than 1 % to S(IV) oxidation in the model.

**Technical Comments**

- The physical properties "mixing ratio" and "concentration" are used as if they were identical. This is not the case! (for details, see http://www.rolf-sander.net/res/vol1kg.pdf) Please check all occurences of the word "concentration" in the main text and check if it should read "mixing ratio" instead.

- page 13, line 366: If Le Breton et al. is still in review, it should be called (2016) not (2015).

- page 16, line 439: The unit "permil" is missing for $\Delta^{17}O$.

---

## Referee Comment (RC2) · A.-L. Norman (Referee) · 30 Jul 2016

This paper represents a very important contribution to the literature on sulfur dioxide oxidation in the remote marine boundary layer and has large implications for contributions to the global sulfate budget from marine biogenic sulfate sources (oxidation of dimethyl sulfide (DMS)) and the role of halogens. It demonstrates for the first time, measurements that constrain the proportion and amount of HOX necessary to be consistent with previously proposed halogen oxidation mechanisms for DMS by von Glascow. It also shows latitudinal bands where the role of HOX may be more important.

I have just a few other comments with respect to the arguments in the manuscript:

[Figure]

1. First, in paragraph 230, the fractional contributions are discussed rather than the concentration. Later in the manuscript, the authors address the concentration but it may be worth mentioning the rationale for describing fractional contribution rather than amount here. 2. Second, dark OH reactions (from nitrogen reactions on aerosol surfaces, e.g. Fuchs et al., 2013 doi:10.1038/ngeo1964) has recently been described as an important night-time oxidation pathway that is typically not considered in chemistry and aerosol models. What would the implications be here and can it be ignored? 3. Third, how realistic is it that S(IV) + HOX results in no pH change in clouds (paragraph 420) and although this is the treatment used in the model, what are the difficulties associated with changing both cloud pH and the fraction of S(IV) + HOX at the same time?

Otherwise the manuscript is in excellent shape and is appropriate for publication. The tables, figures and captions are all appropriate and sufficiently descriptive and the appropriate literature has been cited.

---

## Author Comment (AC1) · 23 Aug 2016

**Responses to Referee #1**

We thank Referee #1 for the helpful comments. Please find our responses below.

**Major issues**

1. I find it confusing that the manuscript switches between cloud droplets and aerosol particles. It looks like measurements of aerosols are compared to model results of cloud chemistry. Is this the case? If yes, please explain why this approach can be valid.

**Response:** This is not the case. Sulfate aerosol in the environment is produced via gas-phase and aqueous-phase chemistry, the latter occurring primarily in clouds. The model is used to distinguish between the different sulfate aerosol production mechanisms (gas, aqueous, heterogeneous) in order to interpret the observations of sulfate isotopes. We have incorporated the following changes to the manuscript to help make this more clear.

**Changes to the manuscript:** Add "The model calculates $\Delta^{17}O$ of bulk sulfate in the model grid box ($\Delta^{17}O_{mod}(nssSO_4^{2-})$) based on the relative importance of each sulfate production mechanism for total sulfate abundance. This is then compared to the $\Delta^{17}O$ measurement of bulk sulfate collected on aerosol filters ($\Delta^{17}O_{obs}(nssSO_4^{2-})$) to investigate sulfate formation mechanisms in the MBL (Sect. 4.2)." to Line 180, in Section 3.

2. In Table 1, you cite Fogelman et al. (1989) and Troy and Margerum (1991) as references for $\Delta^{17}O$ = 0 for the reactions S(IV) + HOX. However, I do not see any isotope chemistry in these papers. How do you derive this zero value? How would your results change if $\Delta^{17}O$ is non-zero for S(IV) + HOX?

**Response:** The Eqs. (2-3) in Fogelman et al. (1989) and Eqs. (5-6) in Troy and Margerum (1991) show that reactions between HOX and $SO_3^{2-}$ occur via nucleophilic attack of $SO_3^{2-}$ onto the X atom of HOX (X = Br or Cl), followed by rapid hydrolysis of $XSO_3^-$. This is shown in Eqs. (1-2) in our paper. The mechanism can be used to determine the resulting $\Delta^{17}O$ value of sulfate. We can see that the three oxygen atoms of $XSO_3^-$ are from $SO_3^{2-}$, which will have a $\Delta^{17}O$ value of 0 ‰ via isotopic exchange of S(IV) with water. Hydrolysis adds one oxygen atom from $H_2O$ to $XSO_3^-$ to form $SO_4^{2-}$. Since all four oxygen atoms of sulfate are derived from water, the resulting $\Delta^{17}O$ value of sulfate will be the same as water, which is 0 ‰ . As long as $SO_3^{2-}$ does not get an oxygen atom directly from HOX to form $SO_4^{2-}$, the $\Delta^{17}O$ of $SO_4^{2-}$ formed via reaction "$SO_3^{2-}$ + HOX" will be 0 ‰ . Liu (2000) suggests the reaction of HOBr with $HSO_3^-$ follows a similar pathway as with $SO_3^{2-}$, i.e. nucleophilic attack of $SO_3^{2-}$ onto

the Br atom of HOBr, followed by rapid hydrolysis of $BrSO_3^-$. Thus, the $\Delta^{17}O$ of $SO_4^{2-}$ formed via reaction "$HSO_3^-$ + HOBr" will also be 0 ‰ . There is no laboratory study on the reaction "$HSO_3^-$ + HOCl", but it is reasonable to assume the same reaction mechanism as for these other pathways leading to $\Delta^{17}O$ = 0 ‰ .

For $\Delta^{17}O_{HOX}(SO_4^{2-})$ to be non-zero, the mechanism for sulfate formation via HOX oxidation would need to follow direct transfer of the oxygen atom of HOX to S(IV). This would lead to $\Delta^{17}O_{HOX}(SO_4^{2-})$ values similar to that from $O_3$ oxidation, and thus HOX oxidation would not resolve the model-measurement discrepancy. However, the above-mentioned laboratory studies all suggest that this is not the case.

We have made the following changes to the manuscript to make it more clear how we determine $\Delta^{17}O_{HOX}(SO_4^{2-})$:

**Changes to the manuscript:** At Line 104, change "Since HOX promotes sulfate formation by adding one oxygen atom from $H_2O$ to sulfate instead of transferring its own oxygen atom (Fogelman et al., 1989; Troy and Margerum, 1991; Yiin and Margerum, 1988), the $\Delta^{17}O$ of sulfate produced from HOX oxidation is expected to be 0 ‰ ." to "Since HOX promotes sulfate formation via "$SO_3^{2-}$+HOX" reactions by adding one oxygen atom from $H_2O$ to sulfate instead of transferring its own oxygen atom (Fogelman et al., 1989; Troy and Margerum, 1991; Yiin and Margerum, 1988), the $\Delta^{17}O$ of sulfate produced from "$SO_3^{2-}$+HOX" reactions is expected to have the same $\Delta^{17}O$ as water (0 ‰ ) (Gonfiantini et al., 1993). Liu (2000) suggests the reaction of HOBr with $HSO_3^-$ follows a similar pathway as with $SO_3^{2-}$ (Eqs. 1-2), resulting in $\Delta^{17}O$ of 0 ‰ for sulfate produced via this reaction. We assume that the reaction of HOCl with $HSO_3^-$ follows a similar pathway as the reaction of HOBr with $HSO_3^-$ (Eqs. 3-4) and produces sulfate with $\Delta^{17}O$ of 0 ‰ . Based on these mechanistic studies, the $\Delta^{17}O$ of sulfate produced from HOX oxidation is expected to be 0 ‰ .".

3. You conclude that a fraction of 33 to 50 % of the sulfate can be produced by about 0.1 pmol/mol HOX. Vogt et al. (1996), however, needed 100 times more HOX (10 pmol/mol)

to achieve a similar fraction (60 %). Why is there such a big difference between these two studies? Can the difference be explained by different rate coefficients that were used? Were different reactions included in these studies? If not, what else could it be?

**Response:** The rate constants used in our calculations are $k_{HOX+SO_3^{2-}} = 2.9 \times 10^9 \, M^{-1} s^{-1}$ and $k_{HOX+HSO_3^-} = 2.0 \times 10^9 \, M^{-1} s^{-1}$. In comparison, the rate constants used in Vogt et al. (1996) are $k_{HOCl+SO_3^{2-}} = k_{HOCl+HSO_3^-} = 7.6 \times 10^8 \, M^{-1} s^{-1}$ and $k_{HOBr+SO_3^{2-}} = k_{HOBr+HSO_3^-} = 5.0 \times 10^9 \, M^{-1} s^{-1}$. Thus, we have used higher rate constants for "S(IV)+HOCl" and lower rate constants for "S(IV)+HOBr". Given the similarity in the overall rate constants, the HOX concentration difference in Vogt et al. (1996) and our study is unlikely to be explained by the difference in rate constants.

The difference of calculated HOX concentration between our study and Vogt et al. (1996) is caused by several reasons described below:

(1) We use a higher Henry's law constant for HOX ($H_{HOX} = 975 \, M \, atm^{-1}$), which will require a lower gas-phase HOX concentration to produce the same amount of sulfate. In Vogt et al. (1996), the Henry's law constant for HOCl ($H_{HOCl}$) and HOBr ($H_{HOBr}$) is 660 $M \, atm^{-1}$ and 93 $M \, atm^{-1}$, respectively. The updated Henry's law constants used in our study requires about an order of magnitude lower concentrations of gas-phase HOX.

(2) The averaged $H_2O_2$ and $O_3$ concentrations ([$H_2O_2$]=0.6 ppt, [$O_3$]=18 ppb) in our studies are lower. In Vogt et al. (1996), [$H_2O_2$] and [$O_3$] are 0.8 ppt and 40 ppb, respectively. Thus, compared to our study, they need higher HOX concentrations to compete with $H_2O_2$ and $O_3$ to produce sulfate.

(3) The HOX concentration of 0.1 ppt in our study is calculated by assuming equilibrium between gaseous and aqueous phase of HOX. The calculation of HOX in Vogt et al. (1996) considers all mass transfer limitations. It is possible that aqueous-phase HOX concentrations are lower than their equilibrium concentrations due to fast chemical

removal in the aqueous-phase. In Vogt et al. (1996), HOBr is removed by both $Cl^-$ and S(IV) in the aqueous phase. If $Cl^-$ concentrations are high enough, it will compete with "S(IV) + HOBr" for loss of HOBr. Von Glasow et al. (2002) has shown that HOBr becomes more important than HOCl in terms of S(IV) oxidation in clouds where LWC is high and $Cl^-$ is diluted.

(4) The HOX concentration of 0.1 ppt calculated in our study is a 24-hour-mean concentration. The HOX concentration in Vogt et al. (1996) is about 10 ppt during the day, but is 2 orders of magnitude lower at night. Thus, the daily mean HOX concentration is on the order of several ppt in Vogt et al. (1996), rather than 10 ppt stated in the comment above.

We have added the following discussion to the manuscript:

**Changes to the manuscript:** Add one paragraph to Line 355: "Uncertainties in our calculated $[HOX]_g$ can originate from (1) uncertainties in the rate constant for reaction between HOX and $HSO_3^-$ ($k_{HOX+HSO_3^-}$), (2) uncertainties in the Henry's law constant for HOX ($H_{HOX}$) and (3) the efficiency of reactive uptake of gas-phase HOX onto cloud droplets that is not accounted for in our assumption of equilibrium of HOX between the gas and aqueous phase.

In comparison, a box-modeling study by Vogt et al. (1996) estimated that daytime-averaged $[HOX]_g$ on the order of 10 ppt is needed to achieve a similar fraction ($\approx 60\%$) of sulfate formed via HOX oxidation. The difference in $[HOX]_g$ between our study and Vogt et al. (1996) is caused by several factors. First, $H_{HOBr}$ in our calculations is an order of magnitude larger than that in Vogt et al. (1996), so that our calculations require an order of magnitude lower $[HOBr]_g$ to produce the same aqueous-phase concentration. Second, $H_2O_2$ and $O_3$ mixing ratios in our calculations (Eq. 9) are lower than those in Vogt et al. (1996) (0.6 ppt vs. 0.8 ppt for $[H_2O_2]$ and 18 ppt vs. 40 ppt for $[O_3]$), so that higher $[HOBr]_g$ is needed in Vogt et al. (1996) to compete with S(IV) oxidation by $H_2O_2$ and $O_3$. Third, in our simple calculation we assume

equilibrium of HOX between the gas- and aqueous-phase while Vogt et al. (1996) considers all mass transfer limitations. Higher $[HOX]_g$ will be calculated if diffusion and subsequent mass accommodation of gas-phase HOX onto the cloud droplets is not fast enough to compensate for the loss of HOX from aqueous-phase chemistry. Fourth, $[HOX]_g$ on the order of 0.1 ppt calculated in our study is a daily-averaged concentration while $[HOX]_g$ on the order of 10 ppt in Vogt et al. (1996) is a daytime-averaged concentration, and hence are not directly comparable. Vogt et al. (1996) has shown that nighttime-averaged $[HOX]_g$ is about two orders of magnitude lower than daytime-averaged $[HOX]_g$."

Delete "In comparison," at Line 356.

**Specific Comments**

1. Abstract: I think it should be mentioned here that uncertainties in reactive halogen concentrations are probably the main reason why halogen chemistry is excluded in large scale models (apart from uncertainties in the reaction rates).

**Response:** Thanks for the comment. We agree that the largest uncertainty in reaction rates is due to the uncertainty in the concentrations of HOX. We have included the following sentence in our abstract.

**Changes to the manuscript:** Change "a production mechanism typically excluded in large scale models due to uncertainties in the reaction rates" to "a production mechanism typically excluded in large scale models due to uncertainties in the reaction rates, which are due mainly to uncertainties in reactive halogen concentrations" at Line 23.

2. page 5, line 131: Here you list several ions which were measured. Unfortunately, the list does not include bromide. Since the reaction HOBr + S(IV) produces bromide, it would be very interesting to know bromide aerosol concentrations. Would it be possible to analyze the samples for bromide?

**Response:** We have measured $Br^-$ concentration ($[Br^-]$) (unit: $ug/m^3$) for the Xue-

Long cruise samples but not for the Malaspina cruise samples. Fig. 1 shows the relationship between our observations of $\Delta^{17}O(nssSO_4^{2-})$ and $Br^-$ concentrations, and the relationship between our calculated $[HOX]_g$ concentrations and observed $Br^-$ concentrations. There is no apparent relationship between $\Delta^{17}O(nssSO_4^{2-})$ and $[Br^-]$ or between $[HOX]_g$ and $[Br^-]$. One would not necessarily expect a relationship between $[Br^-]$ and $[HOX]_g$ because the production of reactive halogens is dependent on additional factors other than $Br^-$, as discussed below.

We have investigated this relationship further using model results from Schmidt et al. (2016). Fig. 2 shows the global distribution of 1st-model-level (<100m) HOBr abundance and $Br^-$ abundance in January 2007 from GEOS-Chem model results in Schmidt et al. (2016). A scatter plot showing the relationship between HOBr and $Br^-$ is shown in Fig. 3. A significant correlation between HOBr and $Br^-$ in the model is not found. This is because the amount of HOBr produced from $Br^-$ and thus the amount of sulfate produced from HOBr depends on other factors in addition to $Br^-$ concentration. These factors include (1) the acidity of aerosols because regeneration of HOBr is acid catalyzed (Fickert et al., 1999), (2) the amount of ultraviolet radiation that photolyzes HOBr, (3) the concentrations of $HO_2$ and $O_3$ that produce HOBr (Eq. 1) and (4) the concentrations of species such as $Cl^-$ and $Br^-$ that act as a sink for HOBr. Thus, $[Br^-]$ alone is not a good proxy for the "HOBr + S(IV)" reaction.

$$2HO_2 + H^+ + 2O_3 + Br^- + h\nu \rightarrow \text{HOBr} + 4O_2 + H_2O \ \ (1)$$

**Changes to the manuscript:** Add "Bromide aerosol concentrations ($[Br^-]$) were also measured for the Xue-Long samples (Supplementary data). There is no relationship between observed $Br^-$ concentration and $\Delta^{17}O(nssSO_4^{2-})$ nor our calculated $[HOX]_g$ (not shown) because factors such as aerosol pH, sunlight and oxidants play an important role in the acid-catalyzed formation of reactive halogens and removal of HOBr (Fickert et al., 1999; Schmidt et al., 2016), Similarly, there is no relationship between $[Br^-]$ and HOBr mixing ratios in the global modeling study by Schmidt et al. (2016) (not shown). Thus, $[Br^-]$ alone is not a good proxy for the "HOBr + S(IV)" reaction." to

Line 135 in Sect. 2.

We have also added the observed $[Br^-]$ to the Supplementary data.

3. page 6, line 173: Metal-catalyzed oxidation of S(IV) is mentioned here. Could you add the information which metals (Fe?, Mn?) are included in the model and how the model calculates their concentrations? I think this information is necessary to understand why this pathway contributes less than 1 % to S(IV) oxidation in the model.

**Response:** Our model includes Fe and Mn for S(IV) oxidation. We have added this information to the manuscript accordingly, which is presented in more detail in Alexander et al. (2009). The reason why metal-catalyzed oxidation pathway is not important in our samples is that our samples are collected in the remote marine boundary layer where trace metals concentrations are low due to the distance from both anthropogenic (coal combustion) and natural (dust) sources of these metals.

**Changes to the manuscript:** Add "The parameterization of the metal-catalyzed S(IV) oxidation pathway is described in detail in Alexander et al. (2009). The trace metals included are Fe and Mn, whose oxidation states Fe(III) and Mn(II) catalyze S(IV) oxidation. Soil-derived Fe ($[Fe]_{soil}$) is scaled to modeled dust concentration as 3.5 % of total dust mass while soil-derived Mn ($[Mn]_{soil}$) is a factor of 50 lower than $[Fe]_{soil}$. Anthropogenic Mn ($[Mn]_{anthro}$) is scaled as 1/300 of primary sulfate concentration while anthropogenic Fe ($[Fe]_{anther}$) is 10 times that of $[Mn]_{anther}$. We assume that 50 % of Mn is dissolved in cloud water as Mn(II) oxidation state. For Fe, we assume that 10 % of $[Fe]_{anther}$ and 1 % of $[Fe]_{soil}$ is dissolved in cloud water. 10 % of the dissolved Fe is in Fe(III) oxidation state during daytime and 90 % at night." to Line 173.

**Technical Comments**

1. The physical properties "mixing ratio" and "concentration" are used as if they were identical. This is not the case! (for details, see http://www.rolf-sander.net/res/vol1kg.pdf) Please check all occurences of the word "concentration" in the main

text and check if it should read "mixing ratio" instead.

**Response:** Thanks for pointing this out. We have updated our manuscript accordingly.

**Changes to the manuscript:** Change "concentration" into "mixing ratio" at Lines 24, 82-89, 338, 351, 354-369, 379, 391-395 and 447-448.

2. page 13, line 366: If Le Breton et al. is still in review, it should be called (2016) not (2015).

**Response:** Thanks for pointing this out. We have updated our manuscript accordingly.

**Changes to the manuscript:** Change "2015" into "2016" at Lines 88-89, 366 and 563.

3. page 16, line 439: The unit "permil" is missing for $\Delta^{17}O$.

**Response:** Thanks for pointing this out. We have updated our manuscript accordingly.

**Changes to the manuscript:** Add " ‰ " to $\Delta^{17}O$ at Line 439.

**Reference**

Gonfiantini, R., Stichler, W., and Rozanski, K.: Standards and intercomparison materials distributed by the International Atomic Energy Agency for stable isotope measurements, in: Reference and intercomparison materials for stable isotopes of light elements: Proceedings of a consultants meeting held in Vienna, 1–3 December 1993, IAEA-TECDOC-825, International Atomic Energy Agency, Vienna, 1993

Schmidt, J. A., Jacob, D. J., Horowitz, H. M., Hu, L., Sherwen, T., Evans, M. J., Liang, Q., Suleiman, R. M., Oram, D. E., Le Breton, M., Percival, C. J., Wang, S., Dix, B., and Volkamer, R.: Modeling the observed tropospheric BrO background: Importance of multiphase chemistry and implications for ozone, OH, and mercury, J. Geophys. Res.-Atmos.,in press, 2016.

Vogt, R., Crutzen, P. J., and Sander, R.: A mechanism for halogen release from sea-salt aerosol in the remote marine boundary layer, Nature, 383, 327–330,

doi:10.1038/383327a0, 1996.

von Glasow, R., Sander, R., Bott, A., and Crutzen, P. J.: Modeling halogen chemistry in the marine boundary layer. 2: Interactions with sulfur and the cloud-covered MBL, J. Geophys. Res., 107D, 4323, 10.1029/2001JD000 943, 2002.
* * *
[Figure]

[Figure]

**Fig. 1.** The $\Delta$17O(nss-sulfate), [HOX]g and [Br-] for each sample from the Xue-Long cruise.

[Figure]

**Fig. 2.** Global distribution of 1st-model-level (<100m) HOBr abundance and Br- abundance in January 2007 from GEOS-Chem model outputs in Schmidt et al. (2016).

[Figure]

[Figure]

**Fig. 3.** The relationship between HOBr and Br- in Fig. 2.

---

## Author Comment (AC2) · 23 Aug 2016

**Responses to Referee #2**

We thank Referee #2 for the helpful comments. Please find our responses below.

1. First, in paragraph 230, the fractional contributions are discussed rather than the concentration. Later in the manuscript, the authors address the concentration but it may be worth mentioning the rationale for describing fractional contribution rather than amount here.

**Response:** Thanks for this suggestion. We separated the "Result" section (Sect. 4) and "Discussion" section (Sect. 5) in the manuscript. For the "Result" section, we

showed only the observations of concentrations and isotopes and primary modeling results (without HOX contribution). Based on these results, we calculated the fractional contribution of sulfate formation from HOX and $[HOX]_g$, which is now described in the "Discussion" section. In paragraph 230, all the fractional contributions shown are obtained directly from the GEOS-Chem model, which does not include the fractional contribution from HOX and $[HOX]_g$ mixing ratio.

**Additional changes to the manuscript:** Add "Based on the modeled fractional contributions to the sulfate burden from each oxidant except HOX, and knowledge about reaction rate constants of "HOX + S(IV)" reactions and Henry's law constants of HOX, we calculate the amount of HOX needed to explain the discrepancy between $\Delta^{17}O_{mod}(nssSO_4^{2-})$ and $\Delta^{17}O_{obs}(nssSO_4^{2-})$ in Sect. 5.2.4." to Line 234 in Section 4.2.

2. Second, dark OH reactions (from nitrogen reactions on aerosol surfaces, e.g. Fuchs et al., 2013 doi:10.1038/ngeo1964) has recently been described as an important nighttime oxidation pathway that is typically not considered in chemistry and aerosol models. What would the implications be here and can it be ignored?

**Response:** Thanks for raising this interesting question. High nighttime OH concentrations have been observed in forests (Faloona et al., 2001) and urban areas (Lu et al., 2014). The mechanism behind this is still unknown. Possible explanations include unimolecular reactions of isoprene-derived peroxy radicals (Fuchs et al., 2013) and an additional ROX production process from VOCs and additional recycling $(RO_2 \rightarrow HO_2 \rightarrow OH)$ (Lu et al., 2014).

In the offline aerosol version of GEOS-Chem that we used in this study, nighttime OH concentration are set equal to zero, so that nighttime oxidation of S(IV) by OH does not occur in the model. However, as the gas-phase reaction "$SO_2$ + OH" is relatively slow compared to aqueous S(IV) oxidation, adding this dark OH production mechanism will be negligible. In our sensitivity study of doubling OH concentrations shown in Sect. 5.2.1, the fraction of sulfate produced by OH oxidation increases only from 20 % to 27

% globally. In addition, we think this dark OH production can be ignored in our study because most of our samples are from remote marine boundary layer where isoprene and VOCs abundance are low.

**Changes to the manuscript:** Add "A diurnal variation as a function of solar zenith angle is applied to OH concentrations and photolytic loss rates of $H_2O_2$ in the model. $NO_3$ is set to be zero during daytime." to Line 172 in Sect. 3.

Add "Thus, doubling OH concentrations has an insignificant effect on $\Delta^{17}O_{mod}(nssSO_4^{2-})$. The nighttime OH concentrations observed in forests and urban areas (Faloona et al., 2001; Lu et al., 2014) should have insignificant effects on our model results as most of our samples are collected in the remote MBL where isoprene and VOCs abundance are low. Our sensitivity study with doubled OH suggests additional nighttime sources of OH are not able to resolve the modeled overestimate of $\Delta^{17}O(nssSO_4^{2-})$ observations." to Line 277 in Sect. 5.2.1.

3. Third, how realistic is it that S(IV) + HOX results in no pH change in clouds (paragraph 420) and although this is the treatment used in the model, what are the difficulties associated with changing both cloud pH and the fraction of S(IV) + HOX at the same time?

**Response:** We do not have "S(IV) + HOX" reaction in the model. Adding "S(IV) + HOX" reaction in the model will probably decrease cloud pH, which will result in decreases in the fractional contribution of $O_3$ relative to $H_2O_2$ ($f_{O3}/f_{H2O2}$) (Line 422). Thus, the HOBr concentration calculated by assuming no change in pH is likely an overestimate (Line 432). We do not have enough information to estimate the magnitude of pH change after adding "S(IV) + HOX" reaction.

For a follow-up study, we are implementing "S(IV) + HOX" reaction in the model. The cloud pH will change in accordance with the additional sulfate produced via "S(IV) + HOX" reaction, as cloud pH in the model is a function of sulfate and other ion concentrations.

**Changes to the manuscript:** Add "The magnitude of the potential decrease in cloud pH can only be obtained after adding the "S(IV) + HOX" reactions in the model, which will be done in a follow-up study." to Line 422.

**Reference**

Faloona, I., Tan, D., Brune, W., Hurst, J., Barket, D., Couch, T.L., Shepson, P., Apel, E., Riemer, D., Thornberry, T. and Carroll, M.A.: Nighttime observations of anomalously high levels of hydroxyl radicals above a deciduous forest canopy, J. Geophys. Res.: Atmospheres, 106(D20), pp.24315-24333,2001.

Fuchs, H., Hofzumahaus, A., Rohrer, F., Bohn, B., Brauers, T., Dorn, H.P., Häseler, R., Holland, F., Kaminski, M., Li, X. and Lu, K.: Experimental evidence for efficient hydroxyl radical regeneration in isoprene oxidation, Nature Geoscience, 6(12), pp.1023-1026, 2013

Lu, K. D., Rohrer, F., Holland, F., Fuchs, H., Brauers, T., Oebel, A., Dlugi, R., Hu, M., Li, X., Lou, S. R., Shao, M., Zhu, T., Wahner, A., Zhang, Y. H., and Hofzumahaus, A.: Nighttime observation and chemistry of HOx in the Pearl River Delta and Beijing in summer 2006, Atmos. Chem. Phys., 14, 4979-4999, doi:10.5194/acp-14-4979-2014, 2014.

---

## Author Comment (AC3) · 23 Aug 2016

The comment was uploaded in the form of a supplement:
http://www.atmos-chem-phys-discuss.net/acp-2016-395/acp-2016-395-AC3-supplement.zip
* * *